# ZeroTuning: Unlocking the Initial Token's Power to Enhance Large Language Models Without Training

**Feijiang Han**[1]* **Xiaodong Yu**[1,2] **Jianheng Tang**[3]
**Delip Rao**[1] **Weihua Du**[4] **Lyle Ungar**[1]

[1]University of Pennsylvania  [2]AMD  [3]Peking University  [4]Carnegie Mellon University

## Abstract

Token-level attention tuning—a class of training-free methods including Post-hoc Attention Steering (PASTA) and Attention Calibration (ACT)—has emerged as a promising approach for improving frozen LLMs via interpretable interventions. However, these methods rely on auxiliary heuristics to identify "important" task-specific tokens, which can introduce bias and limit applicability when token importance is ambiguous or when optimized kernels make attention maps inaccessible. We propose a simpler alternative: intervening only on the initial token (e.g., `<BOS>` in LLaMA). We theoretically show that adding lightweight biases to this token's attention logits systematically shifts and reshapes downstream attention patterns—an effect amplified by its natural role as an attention sink. Empirically, we find that this tuning can improve LLM performance and better elicit pretrained knowledge, with stronger effects in early layers and distinct scaling preferences across attention heads. Building on these findings, we introduce **ZeroTuning**, a training-free method that improves LLM performance by applying head-specific attention adjustments to the initial token, requiring no parameter updates. We present two variants: a supervised mode that calibrates on validation examples, and an unsupervised mode that directly minimizes output entropy. ZeroTuning requires no KV-cache or decoding changes and is kernel-agnostic (works with SDPA and FlashAttention). It requires only four lines of modification to standard `LlamaAttention` code, achieves gains across 15 datasets, and outperforms prior, more complex methods. For example, on Llama-3.1-8B it yields relative improvements of 19.9% on classification, 4.5% on question answering, and 2.1% on dialogue. ZeroTuning also works out of the box with quantized inference and maintains its improvements as context length increases. Our work provides a lightweight tool for inference-time improvement, advancing both optimization and interpretability. Our code and runnable demo are available at `https://anonymous.4open.science/r/ZeroTuning`.

## 1 Introduction

Training-free methods have been widely explored to enhance the performance of Large Language Models (LLMs) at inference time. Among these, **token-level attention tuning** has emerged as a particularly promising direction, offering an interpretable way to steer model behavior by modifying attention distributions without any parameter updates. Unlike fine-tuning (Hu et al., 2021; Dettmers et al., 2023) or prompt engineering (Wei et al., 2022; Wang et al., 2022; **?**), which largely treat LLMs as black boxes, attention tuning provides a transparent mechanism for guiding the model's focus. Methods such as Post-hoc Attention Steering (PASTA (Zhang et al., 2023a), AutoPASTA (Zhang et al., 2024a)) and Attention Calibration (ACT) (Yu et al., 2024) have demonstrated the potential of this approach, in some cases outperforming prompting-based techniques on complex tasks such as open-domain question answering (Zhang et al., 2024a). Similar principles have also been applied to vision-language models to mitigate hallucinations by re-weighting attention toward image tokens (Liu et al., 2024b; Zhu et al., 2024; Wei & Zhang, 2024).

---

*Corresponding author. feijhan@seas.upenn.edu

However, the efficacy of these methods is fundamentally constrained by their reliance on external, often heuristic, mechanisms for identifying task-specific "important" tokens. This dependency not only introduces the risk of bias (e.g., amplifying misleading cues) but also limits applicability when token importance is ambiguous or when optimized attention kernels make attention maps inaccessible. This limitation motivates a fundamental question: *Is it possible to enhance model performance by tuning a universal, task-agnostic token, thereby bypassing the need for fragile, task-specific token identification?*

In this paper, we show that the answer is yes. The key is not added complexity, but rather leveraging a ubiquitous yet often overlooked architectural artifact: the **initial token** (e.g., <BOS> in LLaMA). While its tendency to function as an "attention sink" is well documented (Xiao et al., 2023; Kaul et al., 2024; Gu et al., 2024; Barbero et al., 2025), its potential for performance enhancement has remained largely untapped.

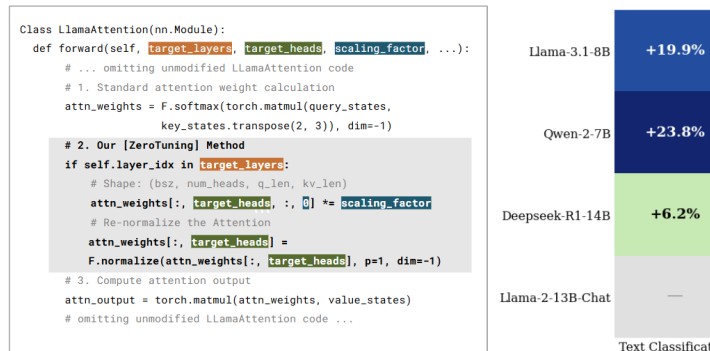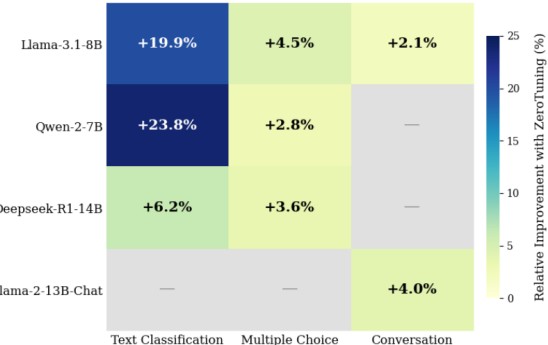

Figure 1: Overview of the **ZeroTuning** method and its effectiveness. **Left:** Our method requires only a few lines of code to scale the initial token's attention within the model's forward pass. **Right:** This simple intervention yields significant and consistent improvements across a variety of LLMs and tasks.

Empirically, we find that steering **only** the initial token's attention can consistently improve LLM performance across diverse tasks, often matching or surpassing prior attention-steering approaches that rely on tuning multiple, task-specific tokens.

To understand **why** this simple intervention works, we proceed in three steps: we first provide a theoretical characterization of how scaling the initial token reshapes attention (Sec. 3.1), then connect the induced redistribution to changes in systematic error patterns and output entropy (Sec. 3.2), and finally localize where the control is most effective by dissecting its impact across layers and heads (Sec. 3.3–3.5)

> **Insights.** Steering the BOS token provides a universal, training-free method to improve LLMs: it modulates attention sharpness, reduces output entropy, mitigates understanding biases, and concentrates the effect in early layers and specific heads.

Building on these findings, we introduce **ZeroTuning** (Figure 1), a simple and training-free method that recalibrates the initial token's attention to boost LLM performance without task-specific token identification (Sec. 3.6). We present two variants: a supervised mode that maximizes accuracy on a labeled validation set, and an unsupervised mode that minimizes output entropy. Across 15 benchmarks, ZeroTuning yields substantial gains on models including Llama-3.1-8B-Instruct, Llama-2-13B-Instruct, Qwen-2-7B, and Deepseek-R1-14B. For example, it improves Llama-3.1-8B-Instruct by a relative 19.9% on classification and 4.5% on question answering, and increases its MT-Bench score from 7.804 to 7.966. The method is robust across long contexts, few-shot settings, quantization, and prompt variations.

## 2 RELATED WORK

Our work is situated at the intersection of two active research areas: inference-time attention tuning and the mechanistic understanding of initial tokens. A growing body of work has shown that modifying token-level attention at inference time can enhance the performance of both LLMs and

VLMs (Yu et al., 2024; Zhang et al., 2023a; Liu et al., 2024b; Wei & Zhang, 2024). However, prevailing methods like PASTA (Zhang et al., 2023a) and Auto-PASTA (Zhang et al., 2024a), which identify and up-weight key tokens, or ACT (Yu et al., 2024), which down-weights non-initial sink tokens, fundamentally rely on heuristics to identify *task-specific* tokens. This reliance limits their universality and introduces potential biases. Concurrently, another line of research has focused on explaining *why* the initial token often becomes an "attention sink" (Xiao et al., 2023), attributing it to architectural biases and its role as a stabilizing anchor (Barbero et al., 2025; Gu et al., 2024). While these studies provide a crucial understanding of *what* the phenomenon is, the question of *how* to actively and elegantly harness it for performance gains remains largely unexplored. Our work bridges this gap. We shift the focus from task-specific token identification to a universal, task-agnostic control point, and move from passive observation of the initial token to a practical tuning framework that leverages its unique properties. Detailed related work is provided in Appendix A.

## 3 UNVEILING THE POWER OF THE INITIAL TOKEN

In this section, we formalize how scaling the initial token's attention reshapes the attention distribution, then empirically demonstrate why this token is uniquely effective and analyze how the effect varies across layers and heads. These findings motivate our ZeroTuning methodology. Unless otherwise specified, all experiments use Llama-3.1-8B-Instruct; see Section 4.1 for setup details.

### 3.1 FORMALIZING THE TUNING PROCESS

In a decoder-only Transformer, autoregressive generation for a sequence $\mathbf{X} = [x_0, x_1, \ldots, x_{T-1}] \in \mathbb{R}^{d \times T}$ relies on causal self-attention. At timestep $T$, the query derived from the final token representation $x_{T-1}$ attends to all preceding token representations (including itself) as keys, producing an attention distribution:

$$\boldsymbol{a} = [a_0, a_1, \ldots, a_{T-1}], \quad \text{where} \quad a_i \geq 0 \quad \text{and} \quad \sum_{i=0}^{T-1} a_i = 1. \tag{1}$$

Here, $a_0$ is the attention score assigned to the initial token, while $a_1, \ldots, a_{T-1}$ correspond to subsequent tokens. To control the influence of $x_0$, we introduce a tuning factor $\gamma > 0$ to scale its attention and re-normalize:

$$a_0' = \frac{\gamma a_0}{D}, \quad a_i' = \frac{a_i}{D} \quad \text{for } i = 1, \ldots, T-1, \tag{2}$$

where the normalization constant $D = \gamma a_0 + \sum_{i=1}^{T-1} a_i = (\gamma - 1)a_0 + 1$.

This rescaling preserves the relative proportions among all non-initial tokens:

$$\frac{a_i'}{\sum_{j=1}^{T-1} a_j'} = \frac{\frac{a_i}{D}}{\sum_{j=1}^{T-1} \frac{a_j}{D}} = \frac{a_i}{\sum_{j=1}^{T-1} a_j}, \quad \text{for } i \geq 1, \tag{3}$$

but compresses or expands their differences as

$$a_i' - a_j' = \frac{a_i - a_j}{D} = \frac{a_i - a_j}{(\gamma - 1)a_0 + 1}, \quad \text{for } i, j \geq 1. \tag{4}$$

Intuitively, $\gamma > 1$ amplifies $a_0$, flattening the remaining distribution, while $\gamma < 1$ suppresses $a_0$, sharpening it. The magnitude of this effect is governed by the initial token's own attention weight, $a_0$. We define $E_{\text{diff},i,j}$ as the change in the attention difference between any two non-initial tokens $i$ and $j$:

$$E_{\text{diff},i,j} = |(a_i' - a_j') - (a_i - a_j)| = |a_i - a_j| \left| \frac{1}{(\gamma - 1)a_0 + 1} - 1 \right| = |a_i - a_j| \frac{|\gamma - 1|a_0}{(\gamma - 1)a_0 + 1}. \tag{5}$$

To analyze how $E_{\text{diff},i,j}$ varies with $a_0$, we take the partial derivative with respect to $a_0$:

$$\frac{\partial E_{\text{diff},i,j}}{\partial a_0} = |a_i - a_j||\gamma - 1| \cdot \frac{1}{((\gamma - 1)a_0 + 1)^2}. \tag{6}$$

Since $|a_i - a_j||\gamma - 1| \geq 0$ and the denominator $((\gamma - 1)a_0 + 1)^2 = D^2 > 0$, the derivative is non-negative. Thus, $E_{\text{diff},i,j}$ is monotonically non-decreasing in $a_0$; in the non-trivial case ($\gamma \neq 1$ and $a_i \neq a_j$), it is strictly increasing. A detailed proof and visualization are in Appendix C.

**Key insight:** the larger the initial token's attention ($a_0$), the more effectively it can act as a lever to control the attention distribution. Since initial tokens are known to be attention sinks (Barbero et al., 2025), they are natural control points for this tuning process.

## 3.2 THE UNIQUE IMPORTANCE OF THE INITIAL TOKEN

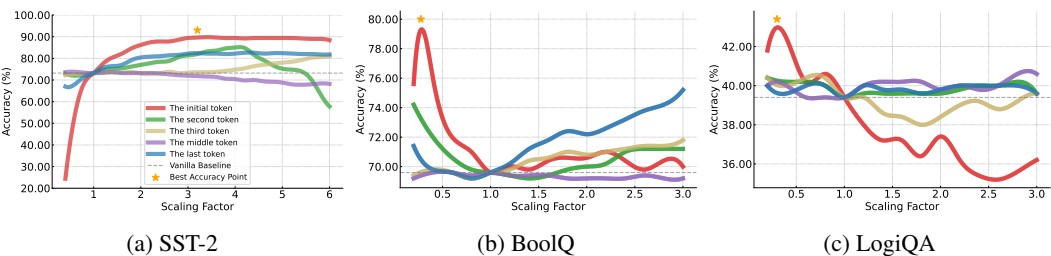

(a) SST-2          (b) BoolQ          (c) LogiQA

Figure 2: Impact of attention scaling factor $\gamma$ on different token positions across three tasks: (a) SST-2, (b) BoolQ, and (c) LogiQA. Modifying the initial token's attention consistently yields significant accuracy improvements, often surpassing adjustments to other tokens.

Given the special role of the initial token, we first investigate a key empirical question: (a) Does tuning its attention positively impact performance on downstream tasks? (b) Is this position more effective and influential than others? To investigate, we conduct a controlled experiment in which we uniformly scale the attention scores of a single token position across all heads and layers using a scaling factor $\gamma$. We evaluate the resulting performance on three downstream tasks: SST-2, BoolQ, and LogiQA. For comparison, we repeat the same procedure for other positions, including the second, third, middle ($\lfloor T/2 \rfloor$), and final tokens. As shown in Figure 2, tuning the initial token consistently yields the largest and most stable gains across tasks. Interestingly, the best direction depends on the task: SST-2 benefits from up-scaling ($\gamma > 1$), while BoolQ and LogiQA improve with down-scaling ($\gamma < 1$).

Previous work has identified the initial token as an *attention sink* that helps prevent over-mixing of information during autoregressive generation (Gu et al., 2024; Barbero et al., 2025). Our results extend this understanding by showing that tuning the initial token reshapes attention over subsequent tokens and improves downstream performance. We interpret this effect from two perspectives.

**1. Correcting Biases.** Tuning counteracts reasoning failures driven by pre-trained attention biases. We observe a consistent pattern: **(a) Up-scaling** ($\gamma > 1$) benefits tasks requiring *holistic context integration*. It flattens attention over non-initial tokens, reducing over-reliance on misleading local cues. For example, in SST-2 (Appendix F), models can fixate on isolated negative words while missing positive context; increasing initial-token attention promotes a more global read and corrects

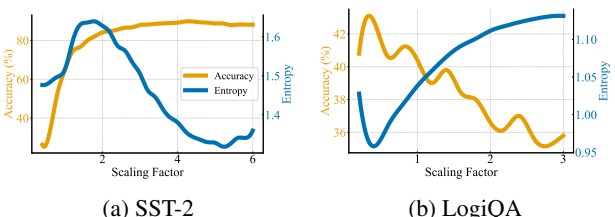

(a) SST-2          (b) LogiQA

Figure 3: A strong inverse correlation between accuracy (**orange**) and next-token prediction entropy (**blue**).

such errors. **(b) Down-scaling** ($\gamma < 1$) benefits tasks requiring *sharp focus* on critical evidence. It increases the relative contrast among non-initial tokens, helping the model localize salient spans in diffuse contexts. In long-context BoolQ, reducing initial-token attention can sharpen focus on the text segment that contains the answer.

**2. Reducing Predictive Uncertainty.** The tuning process can be viewed through the lens of output entropy, a proxy for model uncertainty. As illustrated in Figure 3, a clear inverse correlation emerges: the scaling factor that minimizes entropy consistently aligns with the factor that maxi-

mizes accuracy. This suggests our method better unlocks the model's pretrained knowledge, leading to more confident and correct predictions.

## 3.3 LAYER-WISE ANALYSIS OF INITIAL TOKEN SCALING

To understand how this effect propagates, we apply scaling selectively to different layer groups. Following prior work on layer functionality in Transformers (Jin et al., 2024; Zhang et al., 2024b), we partition the 32 layers of Llama-3.1-8B-Instruct into **shallow** (Layers 1–10), **middle** (Layers 11–21), and **deep** (Layers 22–31). We then scale the initial token within each group and evaluate on six tasks: BoolQ, SST-2, SST-5, MR, LogiQA, and MathQA. Based on Section 3.2, we use $[0, 1]$ for BoolQ and LogiQA, and $[1, 2]$ for the remaining tasks.

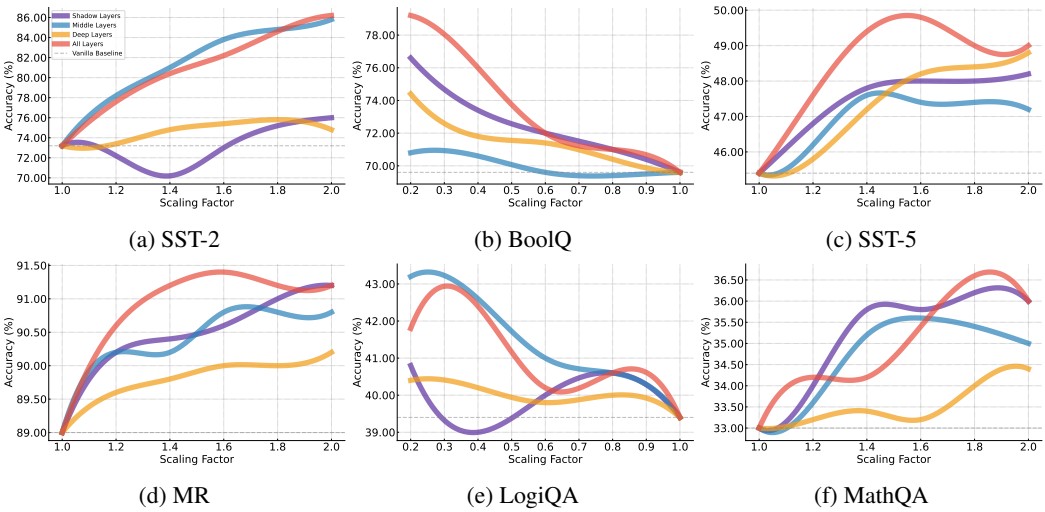

Figure 4: Accuracy trends when scaling the initial token's attention across different layer groups: shallow (Layers 1–10), middle (Layers 11–21), and deep (Layers 22–31). Different depths exhibit a consistent accuracy trend with varying magnitudes.

As shown in Figure 4, scaling induces similar trends across depths, and tuning all layers jointly typically yields the best performance. However, the gains vary in magnitude: tuning shallow and middle layers usually helps more than tuning deep layers.

Prior studies have found that early and middle layers mainly support representation learning and knowledge integration, while deep layers focus on task-specific reasoning over aggregated features (Chen et al., 2024; Jin et al., 2025). Therefore, we argue that the tuning process more effectively reshapes the representational space in shallow and middle layers, promoting better downstream performance and reducing uncertainty.

## 3.4 ANALYZING THE ROLE OF THE INITIAL TOKEN ACROSS ATTENTION HEADS

Unlike layers, attention heads operate in parallel and contribute independently via concatenation, so their responses to initial-token scaling may differ. To test this, we scale the initial token in each head individually with $\gamma = 1.5$ and evaluate on SST-2 and MMLU. For comparison, we also consider (i) no scaling ($\gamma = 1$), (ii) uniform up-scaling across all heads ($\gamma = 1.5$), and (iii) uniform down-scaling across all heads ($\gamma = 0.6$).

As shown in Figure 5, heads respond heterogeneously to initial-token amplification. We call a head *up-effective* if scaling improves performance, and *down-effective* otherwise. The mix of these head types varies across datasets, which helps explain why uniform scaling behaves differently: SST-2 has more up-effective heads and benefits from up-scaling, whereas MMLU has more down-effective heads and thus prefers down-scaling.

These results align with prior studies showing that attention heads specialize into distinct functional roles during pretraining (Zheng et al., 2024b; Guo et al., 2024), such as global retrieval, structural

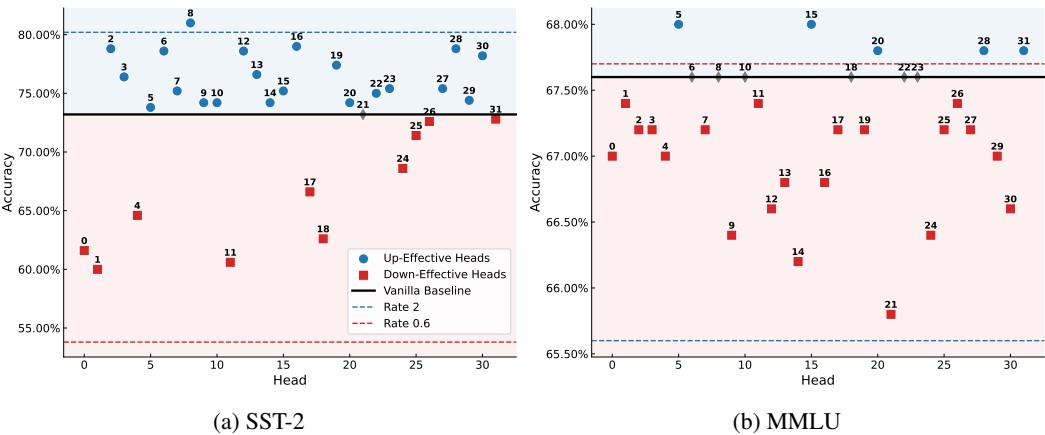

(a) SST-2          (b) MMLU

Figure 5: Accuracy of scaling the initial token's attention in individual heads using $\gamma = 1.5$ across (a) SST-2, (b) BoolQ, (c) MMLU, and (d) MathQA. Results reveal heterogeneous behavior among heads, motivating head-specific tuning strategies.

parsing, option discrimination, and negation sensitivity. We propose that these functional differences may explain the variable impact of initial token attention scaling, with some heads supporting broad global reasoning and others focusing on salient tokens. This interpretation requires further exploration in future work.

## 3.5 EVALUATING HEAD-SPECIFIC TUNING STRATEGIES

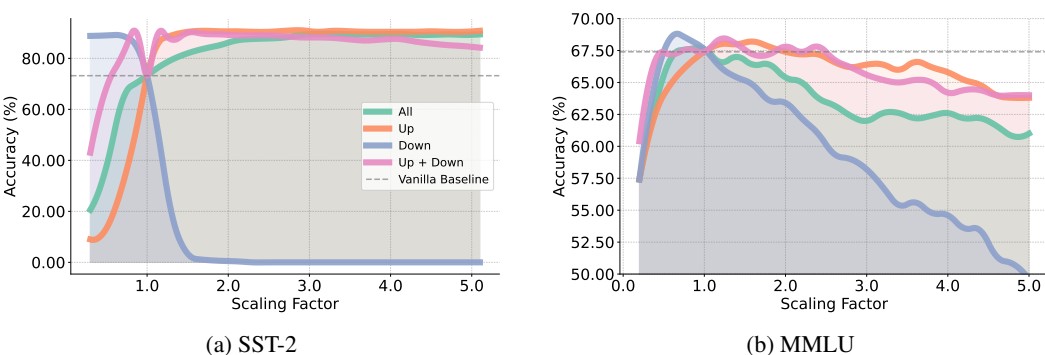

(a) SST-2          (b) MMLU

Figure 6: Accuracy comparison of different tuning strategies on (a) SST-2 and (b) MMLU. Head-specific tuning (UP, DOWN) consistently outperforms uniform scaling, validating the importance of accounting for head-level behavior.

Given the diversity in head responses, we investigate whether head-specific tuning offers greater effectiveness than uniform tuning. Specifically, we compare four strategies: (i) uniform scaling of all heads (ALL), (ii) scaling only up-effective heads (UP), (iii) scaling only down-effective heads (DOWN), and (iv) a hybrid strategy (UP+DOWN) that scales up-effective heads to a fixed optimal value and tunes down-effective ones.

As shown in Figure 6, head-specific tuning (UP, DOWN) yields higher accuracy and faster convergence compared to uniform scaling (ALL). Notably, UP is most effective when $\gamma > 1$, while DOWN excels when $\gamma < 1$. Interestingly, the UP+DOWN strategy does not outperform UP or DOWN individually, possibly due to the concatenative nature of attention heads and suboptimal joint scaling.

## 3.6 ZEROTUNING METHODOLOGY

Building on these findings, we propose **ZeroTuning**, which improves LLM performance by applying head-specific scaling to the initial token without identifying task-specific tokens. ZeroTuning

has three steps: (i) head behavior profiling, (ii) selective rescaling, and (iii) renormalization. For calibration (steps (i)–(ii)), we introduce both supervised and unsupervised variants.

**Supervised Calibration**    Consistent with standard practices in inference-time adaptation (Yu et al., 2024; Zhang et al., 2023a; 2024a), our primary approach utilizes a labeled calibration set (e.g., validation examples) to perform:

1. **Head Behavior Profiling:** Assessing each attention head's sensitivity to the initial token's attention scaling. A head is classified as *up-effective* if increased attention improves accuracy, and *down-effective* otherwise.

2. **Selective Rescaling:** Applying a scaling factor $\gamma$, identified by searching for the value that maximizes accuracy on the calibration set, exclusively to the dominant head type (i.e., the most numerous group).

**Unsupervised Calibration via Entropy Minimization**    To mitigate the reliance on labeled data, we propose a novel unsupervised calibration strategy. Based on our finding that a model's output entropy strongly correlates with its accuracy (Section 3.2), this variant does not require any labeled calibration or validation set. Instead, it identifies the optimal heads and scaling factor $\gamma$ by minimizing the average next-token prediction entropy over a batch of unlabeled inputs. Crucially, the unlabeled inputs can be obtained in two practical ways: (i) in an offline setting, by running Zero-Tuning on a held-out, unlabeled corpus from the same domain; or (ii) in a test-time adaptation style, by performing entropy-based search over the current batch of test-time queries. We provide a more detailed analysis of this unsupervised variant in Appendix E.

Finally, **Renormalization** is shared by both variants: we re-normalize the scaled scores (via softmax) to maintain a valid distribution. For optimized attention implementations (e.g., SDPA, FlashAttention) where direct score modification is infeasible, ZeroTuning applies scaling to the query or key states; Appendix G shows this yields similar effects.

## 4    EXPERIMENTAL RESULTS

### 4.1    EXPERIMENTAL SETUP

Our evaluation includes four recent LLMs with distinct attention implementations (Llama-3.1-8B, Llama-2-13B, Qwen-2-7B, and DeepSeek-R1-14B). We test performance across 15 datasets spanning three categories: Text Classification, Multiple-Choice QA, and Multi-Round Conversation. We benchmark ZeroTuning against three methods: (1) Vanilla inference; (2) ACT (Yu et al., 2024), a sink-token down-scaling method [1]; and (3) Auto-PASTA (Zhang et al., 2024a), an LLM-guided key-token up-scaling method. All experiments are conducted in a zero-shot setting with greedy decoding for fair comparison. For our supervised variant and the baselines, hyperparameters are calibrated on a fixed validation set. A detailed description of all models, datasets, baselines, and implementation specifics is provided in Appendix B.

### 4.2    OVERALL PERFORMANCE OF ZEROTUNING

For a fair and direct comparison with existing supervised baselines, we focus on supervised Zero-Tuning in our main experiments.

**Text Classification**    We first evaluate ZeroTuning on various text classification datasets using different LLMs, as shown in Table 1. Despite tuning only a single token, ZeroTuning consistently outperforms baselines and methods that require tuning more tokens. With Llama-3.1-8B-Instruct, it achieves an average improvement of +11.71% over vanilla, with peaks of +22.00% on SUBJ and +18.40% on SST-2. It outperforms AutoPASTA by an average of 7.71%. On Qwen-2-7B, ZeroTuning gains +13.09%, and on Deepseek-R1-14B, it improves by +4.20%, with a notable increase of +11.20% on TREC.

**Domain-Specific Multiple Choice**    Next, we evaluate ZeroTuning on common domain-specific multiple-choice datasets under various settings, as shown in Table 2. For Llama-3.1-8B-Instruct,

---

[1]Since ACT explicitly manipulates attention maps, we only evaluate it on Llama-3.1-8B-Instruct

Table 1: Performance Comparison of Classification Tasks Across Models. The best performance in each dataset is **bolded** and the ZeroTuning method is highlighted in gray.

| Model | Method | Datasets | | | | | | | |
|---|---|---|---|---|---|---|---|---|---|
| | | SST2 | SST5 | MR | BoolQ | CB | TREC | SUBJ | Avg. |
| Llama-3.1-8B-Instruct | Vanilla | 73.20 | 45.40 | 89.20 | 69.60 | 82.14 | 14.00 | 44.60 | 59.59 |
| | ACT | 85.00 | 43.80 | 90.80 | 58.60 | 82.14 | 15.80 | 44.60 | 60.11 |
| | Auto-PASTA | 89.60 | 47.20 | 91.40 | 72.60 | 83.93 | 16.00 | 45.40 | 63.73 |
| | **ZeroTuning** | **91.60** | **52.00** | **92.00** | **82.40** | **89.29** | **26.20** | **66.60** | **71.44** |
| Qwen-2-7B (SDPA) | Vanilla | 78.80 | 45.40 | 72.40 | 85.00 | 78.50 | 12.60 | 13.00 | 55.10 |
| | ACT | – | – | – | – | – | – | – | – |
| | Auto-PASTA | 89.00 | 47.00 | 77.70 | 85.00 | **89.29** | 14.00 | **57.00** | 65.57 |
| | **ZeroTuning** | **89.60** | **47.20** | **87.40** | **86.40** | 85.71 | **26.60** | 54.40 | **68.19** |
| Deepseek-R1-14B (Flash) | Vanilla | 91.20 | 49.40 | 89.20 | 83.40 | 89.29 | 20.80 | 50.40 | 67.67 |
| | ACT | – | – | – | – | – | – | – | – |
| | Auto-PASTA | 92.00 | **52.20** | 89.80 | 83.40 | **92.86** | 22.60 | 50.40 | 69.04 |
| | **ZeroTuning** | **93.00** | 51.20 | **90.20** | **88.00** | **92.86** | **32.00** | **55.80** | **71.87** |

Table 2: Performance Comparison of Multiple-Choice Tasks Across Models.

| Model | Method | Datasets | | | | | | | |
|---|---|---|---|---|---|---|---|---|---|
| | | MMLU | AQUA | MathQA | LogiQA | CQA | PIQA | ARCC | Avg. |
| Llama-3.1-8B-Instruct | Vanilla | 67.40 | 25.69 | 33.60 | 39.40 | 77.60 | 83.60 | 84.62 | 58.84 |
| | ACT | 67.60 | 29.64 | 33.60 | 38.00 | 77.60 | 83.00 | 84.62 | 59.15 |
| | Auto-PASTA | 67.00 | **31.23** | 35.20 | 40.40 | 78.20 | 84.60 | 84.62 | 60.18 |
| | **ZeroTuning** | **68.80** | 30.43 | **36.60** | **42.80** | **80.40** | **85.40** | **85.95** | **61.48** |
| Qwen-2-7B (SDPA) | Vanilla | 69.80 | 36.76 | 39.20 | 45.00 | 78.80 | 85.20 | 86.96 | 63.10 |
| | ACT | — | — | — | — | — | — | — | — |
| | Auto-PASTA | 69.80 | 39.13 | 39.20 | 45.00 | **82.60** | 85.40 | 86.96 | 64.01 |
| | **ZeroTuning** | **70.40** | **39.92** | **40.20** | **47.40** | 81.80 | **86.20** | **87.96** | **64.84** |
| Deepseek-R1-14B (Flash) | Vanilla | 66.60 | 38.74 | 38.20 | 27.80 | 78.20 | 84.20 | 86.62 | 60.05 |
| | ACT | — | — | — | — | — | — | — | — |
| | Auto-PASTA | 66.60 | 38.74 | 39.40 | 28.20 | 78.20 | 84.40 | 86.62 | 60.31 |
| | **ZeroTuning** | **70.00** | **39.13** | **39.80** | **35.60** | **78.60** | **85.00** | **87.29** | **62.20** |

it increases the average accuracy by +2.64%, with gains of +3.40% on LogiQA and +1.40% on MMLU. Qwen-2-7B gains +1.74%, and Deepseek-R1-14B gains +2.15%, with an outstanding +7.80% on LogiQA.

**Multi-Round Conversation** We further demonstrate ZeroTuning's effectiveness in multi-round conversations using MT-Bench (Zheng et al., 2024a), with results in Table 3. For Llama-3.1-8B-Instruct, ZeroTuning improves the average score by 0.162 points (7.966 vs. 7.804). For Llama-2-13B-Chat, it achieves a 0.266 points gain (6.916 vs. 6.650), showing its effectiveness in interactive settings

Table 3: MT-Bench Performance Scores for Multi-Round Conversation Across Models

| Model | First Turn | Second Turn | Average |
|---|---|---|---|
| gpt-4 | 8.956 | 9.025 | 8.991 |
| **Llama-3.1-8B-ZeroTuning** | **8.294 (+0.029)** | **7.638 (+0.282)** | **7.966 (+0.162)** |
| Gpt-3.5-turbo | 8.075 | 7.813 | 7.944 |
| claude-instant-v1 | 7.800 | 8.013 | 7.906 |
| claude-v1 | 8.150 | 7.650 | 7.900 |
| Llama-3.1-8B-vanilla | 8.265 | 7.353 | 7.804 |
| **Llama-2-13B-Chat-ZeroTuning** | **7.106 (+0.043)** | **6.725 (+0.487)** | **6.916 (+0.266)** |
| Llama-2-13B-Chat-vanilla | 7.063 | 6.238 | 6.650 |

## 4.3 Unsupervised ZeroTuning

We validate a fully unsupervised variant of ZeroTuning, which removes the need for labeled calibration data by minimizing the model's average output entropy. Appendix E provides a detailed empirical validation, including visual comparisons, additional analysis, and a breakdown of error patterns. As shown in Figure 7, this entropy-guided method is highly competitive with its supervised counterpart, extending applicability to label-scarce scenarios.

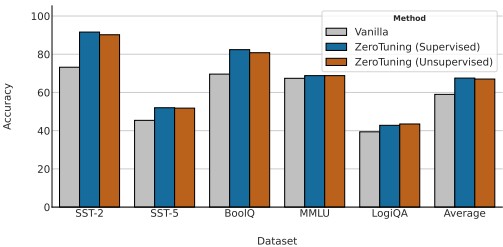

Figure 7: Performance comparison of Vanilla, Supervised, and Unsupervised ZeroTuning on Llama-3.1-8B-Instruct.

## 5 Further Analysis and Key Findings

**Robustness Across Diverse Conditions.** ZeroTuning demonstrates remarkable stability. It maintains strong performance gains even when faced with: **(1)** long contexts padded with irrelevant distractors, where it stabilizes information flow better than the vanilla model (Appendix I); **(2)** few-shot scenarios, where it consistently improves instruction-following and reduces invalid outputs (Appendix J); **(3)** significant prompt variations, including missing or altered instructions (Appendix M); and **(4)** low-precision 4-bit and 8-bit quantization, where it partially mitigates the associated accuracy degradation (Appendix N).

**Practicality and Methodological Choices.** Our method is not only robust but also highly practical. We theoretically and empirically confirm that tuning key states provides a viable, kernel-agnostic alternative to direct attention score manipulation, proving effective in optimized environments like FlashAttention (Appendix G). Furthermore, ZeroTuning is adaptable to resource-constrained settings, delivering gains even with a minimal, search-free scaling approach (Appendix H). We also analyze key methodological choices, showing that tuning a moderate subset of heads (40%–70%) is optimal, providing a clear and efficient configuration (Appendix L).

**Boundaries of Efficacy.** Finally, we analyze the method's boundaries and potential side effects. We quantitatively demonstrate that ZeroTuning excels at correcting a model's uncertain errors but cannot override high-confidence mistakes rooted in flawed pretrained knowledge. We also characterize the negative effects of extreme tuning, which provides a clearer picture of the method's operational limits (Appendix D). This positions our method as a powerful tool for unlocking a model's latent knowledge, rather than a substitute for fine-tuning. Intriguingly, we also find that within a safe operational range, the scaling factor can modulate output diversity in a manner analogous to temperature, but with the unique ability to alter the rank-ordering of logits and thereby correct errors that temperature scaling cannot fix. This aligns with our experiments on temperature tuning, where pure temperature adjustment fails to improve benchmark performance, while ZeroTuning yields consistent gains substantially.

## 6 Further Discussion

### 6.1 Efficient Calibration and Search

In our current implementation, the offline search for the scaling factor $\gamma$ and head selection with Llama-3.1-8B takes roughly 4–5 minutes on a single GPU, and the inference-time overhead is negligible. **We also highlight an interesting empirical observation:** the clear U-shaped entropy curve identified in Appendix E suggests that more efficient search strategies, such as Bayesian optimization or simple rule-based early stopping, could further reduce calibration cost. We view this as a promising direction for future optimization.

### 6.2 ZeroTuning and Supervised Fine-Tuning

Supervised fine-tuning (SFT) remains a strong and versatile way to improve LLMs: by updating parameters, it can simultaneously adjust task knowledge, instruction following, output formatting, and various biases. However, this also means SFT is an indirect optimization over many intertwined behaviors and typically requires more data, compute, and careful hyperparameter tuning.

ZeroTuning takes a complementary approach. Instead of changing parameters, it directly adjusts the attention pattern on the initial token, targeting the specific failure modes identified in our analysis (e.g., format traps and biased attention over context tokens). This makes the intervention more localized and training-free, while still being compatible with SFT.

To study this relationship, we run a controlled experiment on BoolQ where both SFT and supervised ZeroTuning use the same 500 labeled examples. We fine-tune the model with a lightweight LoRA setup (rank $r = 4$, 3 epochs) and compare four configurations:

| Method | BoolQ Accuracy |
|---|---|
| Vanilla | 69.60 |
| Vanilla + SFT (LoRA) | 81.20 |
| Vanilla + ZeroTuning (Ours) | 82.40 |
| Vanilla + SFT + ZeroTuning (Ours) | **83.60** |

In this limited setting, the training-free ZeroTuning slightly outperforms our small-scale LoRA SFT, and applying ZeroTuning on top of the SFT model yields the best performance. While stronger fine-tuning setups (more data, higher rank, more epochs) may implicitly learn to adjust the initial-token attention in a similar way, we hope our observations provide a useful lens for understanding how SFT optimizes attention patterns and inspire future work on combining training-free and training-based approaches.

## 7 CONCLUSION

This work provides a systematic analysis of how scaling the initial token's attention affects a Transformer's attention distribution and downstream accuracy. Building on this analysis, we introduce **ZeroTuning**, a training-free inference-time method that calibrates a single, task-agnostic token. Across a wide range of tasks and models, ZeroTuning improves performance over vanilla inference and is competitive with (and often stronger than) prior inference-time tuning baselines that require manipulating additional tokens or task-specific heuristics. ZeroTuning supports both supervised calibration (via a small labeled set) and unsupervised calibration (via entropy minimization), and it can be implemented across different optimized attention kernels. Overall, our results highlight the initial token as a simple but effective control point for lightweight, deployment-friendly model adaptation.

## ACKNOWLEDGMENTS

We extend special thanks to **Prof. Surbhi Goel** from the University of Pennsylvania for her valuable advice on the theoretical aspects of this research.

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

# A RELATED WORK

## A.1 TOKEN-LEVEL ATTENTION TUNING

Token-level attention tuning typically aims to increase attention to critical input tokens or decrease attention to less informative tokens. Lu et al. (2021) proposes a mask perturbation method to adjust attention weights for key tokens, thereby improving translation quality. Zhang et al. (2023a) introduce PASTA, which allows manual designation of important tokens during inference. This is extended by AutoPASTA (Zhang et al., 2024a), which uses LLMs to autonomously identify salient tokens and increase attention to them. In contrast, ACT (Yu et al., 2024) reduces attention to semantically trivial sink tokens and redirects it to meaningful content. Similar strategies have been applied to VLMs to mitigate hallucinations. PAI (Liu et al., 2024b) enhances attention to image tokens at inference time to counteract text-dominant bias. IBD (Zhu et al., 2024) and OPERA (Wei & Zhang, 2024) further refine this idea by prioritizing visual information or penalizing overconfident summary tokens. While effective, these methods depend on identifying task-specific tokens, which may introduce bias (e.g., overemphasizing misleading tokens) and limit applicability when token importance is unclear or attention maps are unavailable. In contrast, our method focuses on a task-invariant initial token, removing the need for costly token identification, and can be easily applied by tuning key states.

## A.2 THE MAGIC OF THE INITIAL TOKEN

Recent studies highlight the significance of the initial token, especially through the lens of the *attention sink* phenomenon, where it draws substantial attention despite low semantic content. Xiao et al. (2023) show that preserving such tokens is critical for maintaining performance in sliding window attention. Kaul et al. (2024) attribute this effect to softmax normalization and causal masking, while Gu et al. (2024) and Barbero et al. (2025) identify architectural biases that amplify attention to the initial token, including key-query alignment and LayerNorm effects. Functionally, the initial token is hypothesized to serve as a stabilizing "no-op" anchor, enhancing robustness to prompt variations (Barbero et al., 2025). It has been leveraged in applications such as long-context modeling (Zhang et al., 2023b; Xiao et al., 2023), but also poses challenges for quantization due to its high attention weight (Dettmers et al., 2023; Liu et al., 2024a). While previous work has identified the structural and functional importance of the initial token, its potential as a target for attention tuning remains underexplored. In this work, we provide a detailed analysis of attention tuning of the initial token across layers and heads, demonstrating its consistent influence across different tasks. Our approach bridges the gap between the these lines of research by proposing a novel method that advances interpretable attention tuning.

# B DETAILED EXPERIMENTAL SETUP

**Models, Tasks, and Datasets.** Models: We evaluate ZeroTuning on four LLMs with distinct attention implementations: Llama-3.1-8B-Instruct (Grattafiori et al., 2024) and Llama-2-13B-Chat (Touvron et al., 2023) with eager attention, Qwen-2-7B (Yang et al., 2024) with SDPA attention, and DeepSeek-R1-14B (DeepSeek-AI et al., 2025) with Flash attention.[2] Tasks and Datasets: Our experiments encompass three task types across 15 datasets: (1) Text Classification and Reasoning, including SST-2 (binary sentiment classification) (Socher et al., 2013), SST-5 (fine-grained sentiment analysis) (Socher et al., 2013), MR (movie review polarity detection) (Pang & Lee, 2005), SUBJ (subjectivity classification) (Pang & Lee, 2004), TREC (question type classification) (Li & Roth, 2002), CB (commitment detection) (De Marneffe et al., 2019), and BoolQ (boolean question answering) (Clark et al., 2019); (2) Domain-Specific Multiple-Choice, including MMLU (cross-domain knowledge testing) (Hendrycks et al., 2020), AQUA (math word problems) (Zheng et al., 2024a), MathQA (algebraic reasoning) (Amini et al., 2019), LogiQA (logical reasoning) (Liu et al., 2023), CQA (commonsense reasoning) (Talmor et al., 2018), PIQA (physical commonsense QA) (Bisk et al., 2020), and ARCC (scientific reasoning) (Clark et al., 2018); and (3) Multi-Round Conversation, using MT-Bench (Zheng et al., 2024a).

---

[2] Eager, SDPA, and Flash are official attention implementations in modern Transformer libraries. Eager computes the full attention map; SDPA uses PyTorch's efficient API to select the optimal implementation; Flash relies on fused CUDA kernels from the FlashAttention library.

**Baselines and Evaluation Metrics.** Baselines: We benchmark ZeroTuning against three baselines: (1) vanilla inference, which performs standard inference without any modifications; (2) ACT (Yu et al., 2024), which identifies none-initial sink tokens using an attention score threshold and reduces their attention weights; and (3) Auto-PASTA (Zhang et al., 2024a), which leverages an LLM to locate important tokens and enhance their attention weights. Evaluation Metrics: We assess performance using accuracy for text classification and multiple-choice tasks. For the multi-round conversation task, we report average quality scores as evaluated by GPT-4, following the methodology outlined in Zheng et al. (2024a).

**Implementation Details.** All experiments are implemented in PyTorch using the Hugging Face Transformers library. We use a zero-shot setting with greedy decoding for consistency across all methods. For our supervised variant and the baselines, we use a fixed validation set of 500 randomly selected samples (seed 42) for calibration. For ZeroTuning, we tune the top 40% of identified heads unless otherwise specified. For ACT, we use the official hyperparameter ($\beta = 0.4$), and since it requires explicit attention maps, we only evaluate it on Llama-3.1-8B-Instruct. Prompts for all tasks and baselines are detailed in Appendix O.

## C  THEORETICAL ANALYSIS OF TUNING EFFICACY VIA THE INITIAL TOKEN

This appendix provides a formal proof for the claim made in Section 3.1: that the tuning effect's magnitude is governed by the initial token's attention weight $a_0$.

**Proposition 1.** *For any given scaling factor $\gamma \neq 1$ and any two non-initial tokens $i, j \geq 1$ with unequal initial attention weights ($a_i \neq a_j$), the magnitude of the tuning effect on their attention difference is a strictly monotonically increasing function of the initial token's attention weight, $a_0$.*

We aim to show that the tuning effect, $E_{\text{diff},i,j}$, is a monotonically increasing function of $a_0$ by proving its partial derivative with respect to $a_0$ is positive. Recall the definition of the effect magnitude from Eq. equation 5:

$$E_{\text{diff},i,j}(a_0) = |a_i - a_j| \frac{|\gamma - 1| a_0}{(\gamma - 1) a_0 + 1}. \tag{7}$$

Taking the partial derivative of $E_{\text{diff}}$ with respect to $a_0$, we treat the term $|a_i - a_j||\gamma - 1|$ as a constant factor:

$$\frac{\partial E_{\text{diff},i,j}}{\partial a_0} = |a_i - a_j||\gamma - 1| \cdot \frac{\partial}{\partial a_0} \left( \frac{a_0}{(\gamma - 1) a_0 + 1} \right) \tag{8}$$

$$= |a_i - a_j||\gamma - 1| \cdot \frac{1 \cdot ((\gamma - 1) a_0 + 1) - a_0 \cdot (\gamma - 1)}{((\gamma - 1) a_0 + 1)^2} \tag{9}$$

$$= |a_i - a_j||\gamma - 1| \cdot \frac{1}{((\gamma - 1) a_0 + 1)^2}. \tag{10}$$

The term $|a_i - a_j||\gamma - 1|$ is non-negative. The denominator, $((\gamma - 1) a_0 + 1)^2 = D^2$, is the square of the normalization constant and is strictly positive for any valid probability distribution. Therefore, the derivative $\frac{\partial E_{\text{diff},i,j}}{\partial a_0} \geq 0$.

Furthermore, for any non-trivial case where the tuning factor is active ($\gamma \neq 1$) and the attention weights are not uniform ($a_i \neq a_j$ for some $i, j$), the derivative is strictly positive. This proves that $E_{\text{diff}}$ is a strictly monotonically increasing function of $a_0$. Consequently, a larger initial attention weight provides a more powerful lever for modulating the attention distribution. This theoretical result is visually corroborated by Figure 8.

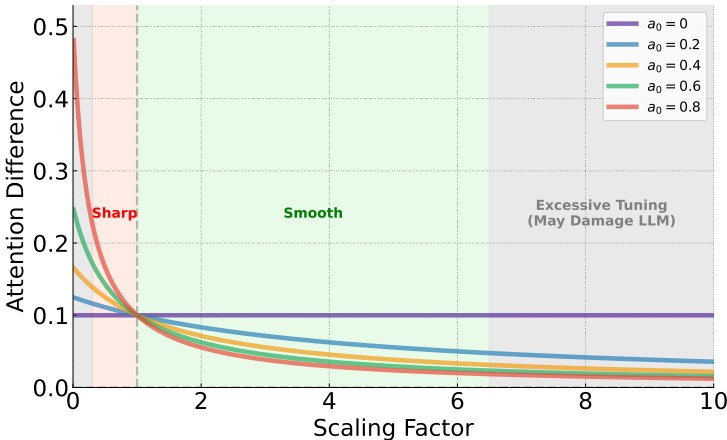

Figure 8: Visualization of the tuning effect on attention differences $|a'_i - a'_j|$ as a function of the scaling factor $\gamma$ for various initial attention weights $a_0 \in \{0, 0.2, 0.4, 0.6, 0.8\}$. The plot demonstrates that a higher initial attention weight $a_0$ (e.g., the red curve) leads to a significantly stronger response to changes in $\gamma$. We identify three primary regimes: **sharpening** ($\gamma < 1$), where attention differences are amplified; **smoothing** ($\gamma > 1$), where differences are diminished; and regions of **excessive tuning** (e.g., $\gamma \to 0$ or $\gamma \gg 1$), which may degrade performance.

## D    DEEPER ANALYSIS OF POTENTIAL FAILURES AND NEGATIVE EFFECTS

To provide a comprehensive understanding of ZeroTuning, we analyze its operational boundaries and potential negative side effects when pushed to its limits.

**Boundaries of Efficacy: Unlocking Latent Knowledge vs. Correcting Factual Errors.**   Our analysis reveals a key insight into ZeroTuning's mechanism: it primarily unlocks and disambiguates a model's latent knowledge, rather than correcting deeply ingrained factual errors. To test this, we quantitatively analyzed ZeroTuning's corrective power as a function of the vanilla model's initial prediction confidence. We partitioned the set of SST2 incorrect predictions into "uncertain errors" (vanilla softmax confidence < threshold) and "certain errors" (confidence ≥ threshold) and evaluated our method's performance on each group.

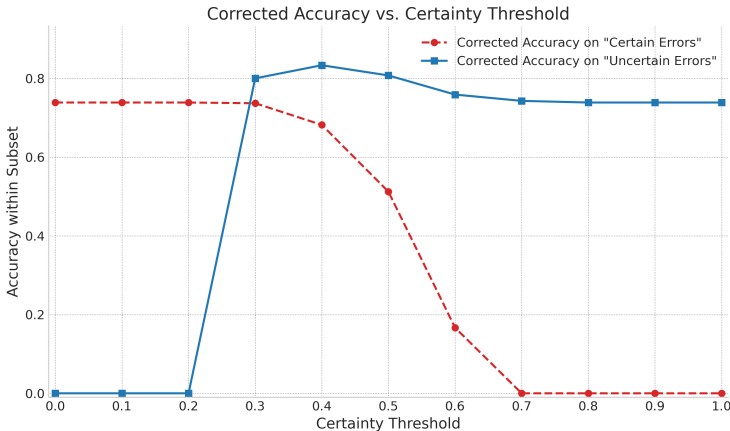

Figure 9: Corrected accuracy on initially incorrect samples as a function of the vanilla model's confidence (certainty threshold).

The results, shown in Figure 9, provide strong empirical validation. ZeroTuning is highly effective on the uncertain error population (blue line), correcting over 80% of mistakes where the vanilla

model's confidence was below 0.5. Conversely, its ability to fix certain errors (red line) decays sharply as the base model's confidence increases, dropping to near-zero on predictions where the model was already confidently wrong. This confirms a clear operational boundary: ZeroTuning excels at resolving low-confidence mistakes by refining the model's focus, but it is not designed to overwrite high-confidence knowledge learned during pretraining.

This finding positions ZeroTuning not as a replacement for fine-tuning, but as a powerful, complementary inference-time technique. It assists a model in better leveraging its existing, albeit sometimes uncertain, knowledge. The potential for synergistic interaction between ZeroTuning and parameter-efficient fine-tuning methods like LoRA remains a promising avenue for future research.

**Negative Effects of Extreme Tuning.** We also investigated the effects of applying extreme scaling factors ($\gamma$), far outside the optimal range. These experiments reveal predictable failure modes that further illuminate the role of the initial token:

- **Overly-suppressed attention** ($\gamma \to 0$)**:** When the initial token's attention is excessively reduced, the model's output often becomes degenerative. We observe a tendency for the model to enter repetitive loops, outputting a single answer (e.g., "True") without any of the semantic elaboration or reasoning present in the vanilla output. This suggests that a minimal level of attention to the "sink" token is necessary to maintain generative stability.
- **Overly-amplified attention** ($\gamma \gg 1$)**:** Conversely, when the initial token's attention is excessively high, it can disrupt the model's ability to follow complex instructions. By absorbing too much of the attention budget, the initial token appears to prevent other, more task-relevant tokens from receiving the focus they need, leading to incomplete or non-compliant answers.

Interestingly, within a reasonable range, moderate tuning of the initial token's attention can produce effects analogous to adjusting the temperature parameter in decoding. It can modulate the diversity of the output, encouraging the model to explore different perspectives or generate more varied responses, without the repetitive downsides of extreme scaling.

However, our method is fundamentally more powerful for error correction. Temperature scaling acts on the final logits $z$ just before the softmax, calculating the probability of the $i$-th token as $p_i = \text{softmax}(z_i/T)$. Since dividing by a positive temperature $T$ does not change the relative order of the logits (i.e., $\arg\max_i(z_i) = \arg\max_i(z_i/T)$), temperature scaling cannot alter the outcome of greedy decoding. In contrast, ZeroTuning operates at the attention level, optimizing the model's internal representations. This process produces an entirely new set of output logits, $z'$, which can have a different rank ordering. It is therefore possible for the originally predicted token $\arg\max_i(z_i)$ to be incorrect, while the new prediction $\arg\max_i(z'_i)$ becomes correct, enabling error correction.

## E   UNSUPERVISED ZEROTUNING: ANALYSIS AND RESULTS

This section details the unsupervised variant of ZeroTuning, which eliminates the need for a labeled calibration set by leveraging the model's output entropy as a proxy for performance.

We begin with a visual analysis to establish the core principle behind this approach. Figure 10 plots three key metrics against the attention scaling factor $\gamma$: the average next-token entropy, its logarithm, and the final task accuracy. The plots reveal a compelling and consistent pattern across all datasets. The average entropy curve (left subplot) exhibits a distinct U-shape, identifying a clear scaling factor that minimizes the model's predictive uncertainty. Critically, the trough of these entropy curves aligns remarkably well with the peak, or a near-peak plateau, in the accuracy curve (right). This strong visual correlation provides powerful evidence that minimizing entropy can serve as a robust, unsupervised signal for identifying a high-performance region for $\gamma$.

Guided by this insight, we quantify the effectiveness of an unsupervised approach where we select the $\gamma$ that minimizes entropy on the unlabeled test set. Table 4 compares its performance against the vanilla baseline and the supervised ZeroTuning variant. The results demonstrate that the unsupervised method is remarkably effective, achieving an average score of 67.02—highly competitive with the supervised result of 67.52 and a substantial improvement over the vanilla baseline's 59.00. Notably, on LogiQA, the entropy-guided method even slightly outperforms its supervised counterpart.

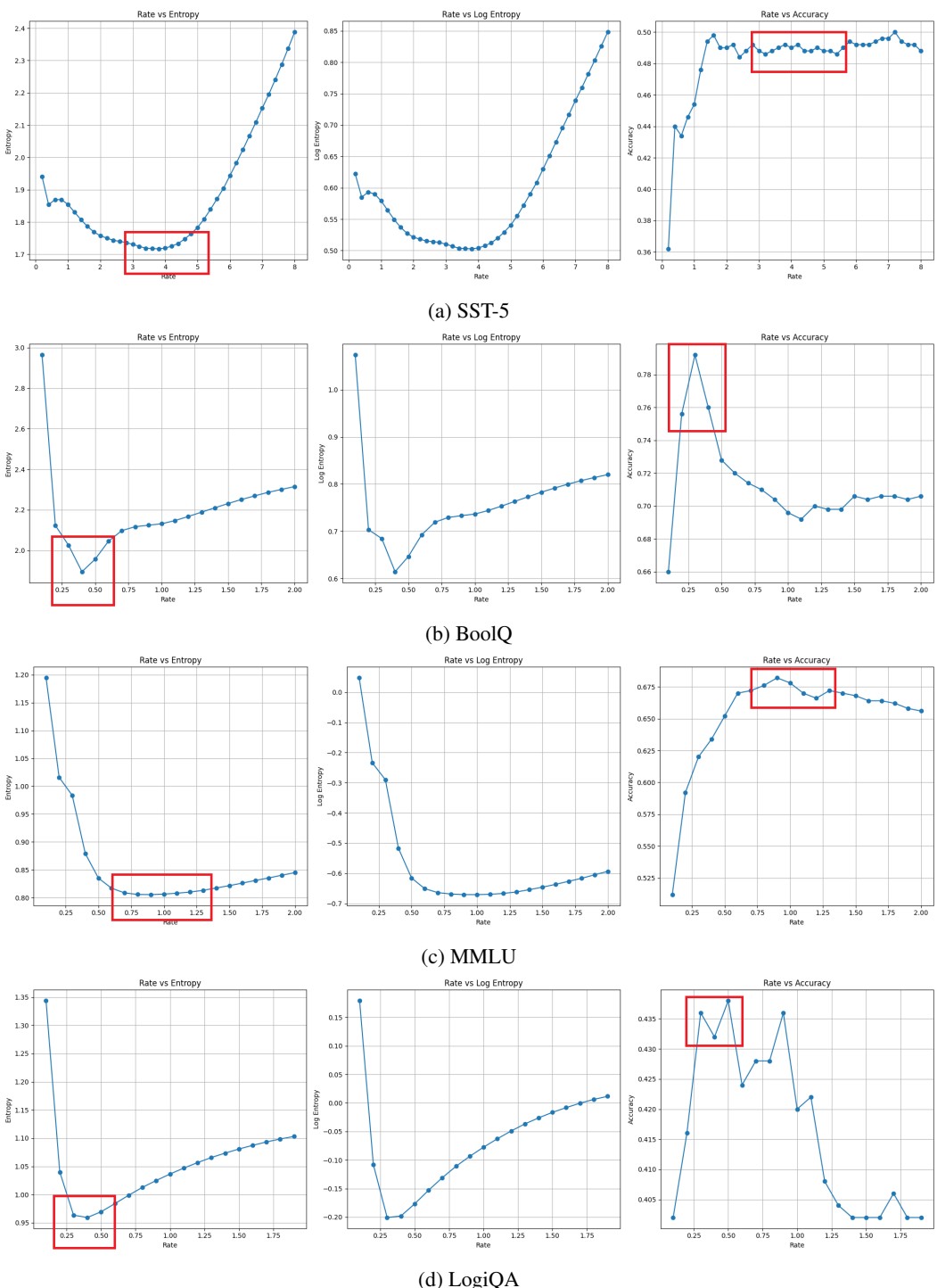

Figure 10: Visualizing the relationship between the scaling factor $\gamma$ and three key metrics: average entropy (left), log-entropy (middle), and task accuracy (right). Across diverse datasets, the entropy minimum consistently aligns with a region of high accuracy, validating entropy as a strong signal for unsupervised tuning.

This quantitative validation confirms that unsupervised ZeroTuning is a powerful and practical alternative, transforming our method into a versatile tool that can be deployed without any task-specific labeled data.

Table 4: Performance comparison of Vanilla, Supervised, and Unsupervised ZeroTuning on Llama-3.1-8B-Instruct. The best performance in each column is **bolded**.

| Method | SST-2 | SST-5 | BoolQ | MMLU | LogiQA | Avg. |
|---|---|---|---|---|---|---|
| Vanilla | 73.20 | 45.40 | 69.60 | 67.40 | 39.40 | 59.00 |
| ZeroTuning (Supervised) | **91.60** | **52.00** | **82.40** | 68.80 | 42.80 | **67.52** |
| ZeroTuning (Unsupervised) | 90.20 | 51.80 | 80.80 | 68.80 | **43.40** | 67.02 |

## E.1 DEEPER ERROR ANALYSIS FOR UNSUPERVISED ZEROTUNING

A key finding of our analysis is that while minimizing entropy on individual samples can be misleading, minimizing the *average* entropy across a dataset robustly identifies an optimal tuning parameter. This suggests Unsupervised ZeroTuning corrects for systemic, dataset-level biases rather than isolated prediction errors. To understand this phenomenon, we analyze how the method behaves on different error types for SST2.

**The Unreliable Entropy Landscape of Uncertain Samples.** We first examine "uncertain" samples, where the vanilla model's confidence in its top-choice token is low (e.g., $p_{top} < 0.5$). For these samples, the entropy landscape is often deceptive and contains two primary "traps":

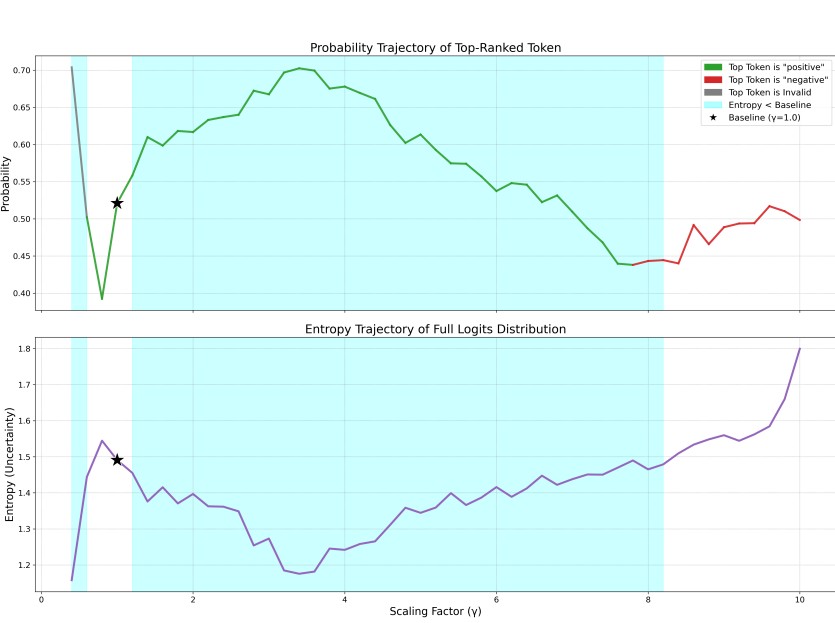

Figure 11: An uncertain sample where the correct answer is "negative". The vanilla model incorrectly outputs "positive". For $\gamma < 1$, the model outputs an invalid format. For $\gamma > 1$, it first amplifies the probability of the initial incorrect answer before flipping to the correct one. The entropy minima are misleadingly located in the regions of the invalid and incorrect answers.

1. **The Format Trap ($\gamma < 1$):** When reducing the initial token's attention, the model enters a "local-focused" mode. For uncertain samples, this can amplify attention on unintended tokens, leading to a violation of task instructions. In this mode, the model often outputs tokens outside the constrained answer space, such as "neutral" or "The", instead of the required "positive" or "negative". As $\gamma$ decreases below 1, the model's confidence in this wrongly formatted token can increase, creating a misleading drop in entropy.

2. **The Bias Amplification Trap ($\gamma > 1$):** When increasing the initial token's attention, the model enters a "global-integrative" mode. For an initially incorrect, uncertain sample, this

Figure 12: A more uncertain sample where the correct answer is "positive". The vanilla model incorrectly outputs "neutral" (an invalid format). As $\gamma$ increases, the model first transitions to a valid but incorrect answer ("negative") before finally flipping to the correct answer ("positive"). The entropy landscape exhibits multiple local minima corresponding to the invalid, incorrect, and correct answers, making direct judgment based on the global minimum unreliable.

often induces a "competing peaks" phenomenon. Both Figure 11 and 12 illustrate cases where the model first amplifies an existing bias, leading to a deep entropy well corresponding to an incorrect answer. Only with further increases in $\alpha$ does the model's interpretation "flip" to the correct one. Figure 12 shows a more complex cascade: after overcoming the format trap, the model first falls into a bias trap (incorrectly predicting "negative" due to the word "not") before finally settling on the correct answer ("positive"). In both cases, if the global entropy minimum is sought, it may lock onto an amplified bias or a formatting error, creating a trap.

**The Stabilizing Behavior of Certain Samples.** In contrast, "certain" samples (where $p_{top} \geq 0.5$) exhibit more predictable behavior and act as a stabilizing force during average entropy minimization. We identify two sub-types:

- **Recalcitrant Certain Errors:** As shown in Figure 13, if the model is confidently wrong but its answer is *within the valid format* (e.g., predicting "negative" for a positive sentence), its semantic conviction is strong. In this case, modulating $\gamma$ reinforces this conviction, leading to a stable, uni-modal probability peak. The error is not corrected. This suggests the model's relevant pretrained knowledge is already strongly activated, albeit towards an incorrect conclusion. These samples act as a stable "ballast" in the collective average.

- **Correctable Certain Errors:** As shown in Figure 14, if the model is confidently wrong because it produced an *out-of-format* token (e.g., "neutral"), the error is rooted in a misunderstanding of task constraints/instruction, not deep semantic conviction. Here, the model's pretrained knowledge is "locked". Modulating $\gamma$ (specifically, increasing it) helps the model refocus on the instructions, "unlocking" its latent knowledge and guiding it to the correct, in-format answer. In these cases, the entropy minimum correctly corresponds to the right answer.

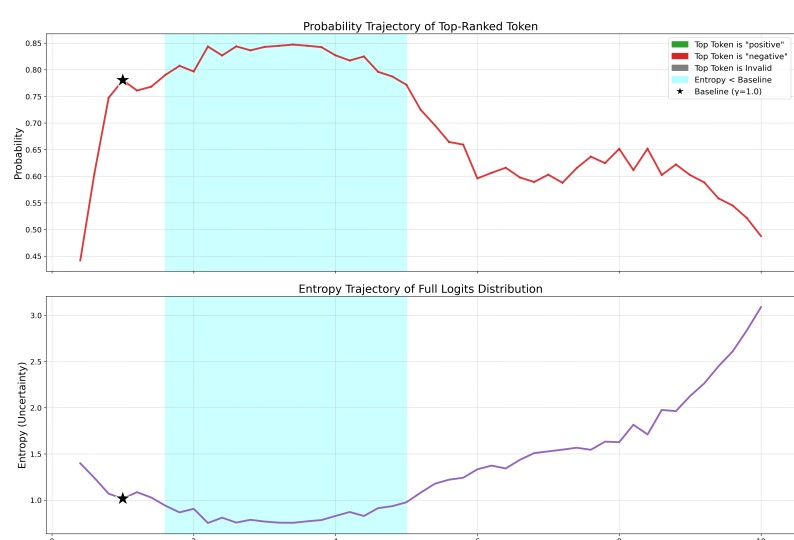

Figure 13: A **recalcitrant** certain sample. The vanilla model (⋆) predicts an incorrect but in-format answer ("negative") with high confidence. Modulating $\gamma$ only reinforces this conviction without an answer flip.

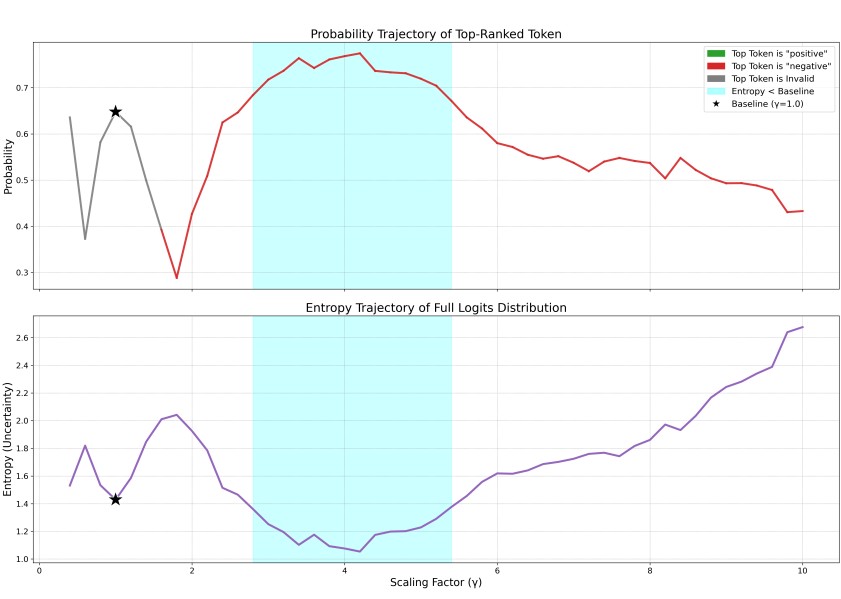

Figure 14: A **correctable** certain sample. The vanilla model (⋆) predicts an out-of-format answer ("neutral") with high confidence. Increasing $\alpha$ guides the model to overcome the format error and output the correct answer ("negative").

**Why Collective Signal Works Better?**   The visualizations in Figure 15 reveal a key insight: while individual uncertain samples exhibit highly irregular and sometimes multi-peaked probability/accuracy trajectories as $\gamma$ varies, the aggregate signal constructed from many samples becomes smooth, stable, and strongly aligned with true accuracy improvement.

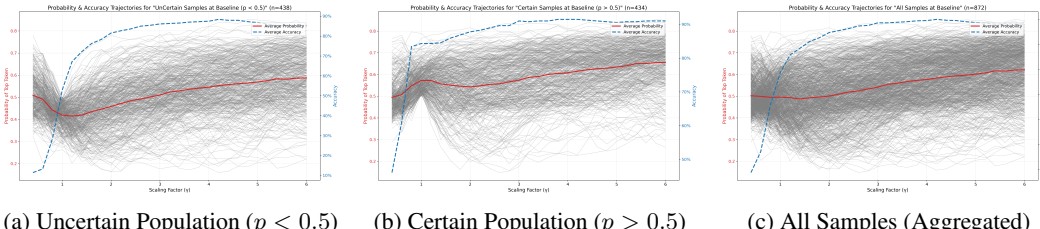

(a) Uncertain Population ($p < 0.5$)     (b) Certain Population ($p > 0.5$)     (c) All Samples (Aggregated)

Figure 15: Population-level probability and accuracy trajectories. Gray lines represent individual sample trajectories. Red: average probability of the top token. Blue: average accuracy.

We observe two complementary mechanisms:

- **Statistical Cancellation for $\gamma < 1$.** Uncertain samples tend to fall into the format trap and display artificially low entropy for invalid outputs, whereas certain samples consistently lose confidence when $\gamma$ decreases. When averaged, these opposing trends cancel out, producing a flat and stable population-level curve. This prevents the collective entropy signal from being dominated by pathological samples.

- **Signal Reinforcement for $\gamma > 1$.** In this region, both populations—despite their very different behaviors at the individual level—exhibit a shared upward trend in accuracy as $\gamma$ increases. This alignment creates a resonant effect: the aggregated probability and accuracy curves become smoother and strongly monotonic, reliably pointing toward the globally beneficial regime of $\gamma > 1$.

Together, these effects explain why Unsupervised ZeroTuning succeeds when optimizing the average entropy. The population average filters out idiosyncratic traps (format errors, bias amplifications) and amplifies the underlying structural signal that reflects the model's latent reasoning and instruction-following capabilities. This makes the minimum of the average-entropy curve over $\gamma$ a trustworthy indicator of the optimal tuning direction.

## F LLM OUTPUT EXAMPLE

### F.1 EXAMPLES OF SST-2 DATASET

**Example 1**:

> Classify the sentence into one of the following sentiments: positive or negative.
> Sentence: "hardly a masterpiece, but it introduces viewers to a good charitable enterprise and some interesting real people."
> Sentiment:
> **Vanilla Output**: negative. Explanation: The sentence describes the movie as "hardly a masterpiece", which is a negative description. The positive aspects mentioned in the sentence are secondary to the negative description. Therefore, the overall sentiment of the sentence is negative.
> **ZeroTuning Output** ($\gamma = 4$): positive. Explanation: The word "good" in the sentence has a positive connotation, and the word "interesting" also has a positive connotation. This indicates that the overall sentiment of the sentence is positive. The word "hardly" is a negative word, but in this context, it is used to express a mild criticism, so it does not dominate the overall sentiment of the sentence. Therefore, the sentence is classified as having a positive sentiment.
> **Ground Truth**: positive

**Example 2**:

> Classify the sentence into one of the following sentiments: positive or negative.
> Sentence: "generally, clockstoppers will fulfill your wildest fantasies about being

a different kind of time traveler, while happily killing 94 minutes."
Sentiment:
**Vanilla Output**: negative. Explanation: Although the sentence mentions "wildest fantasies," which can be interpreted as positive, the overall tone is negative due to the phrase "happily killing 94 minutes." This phrase implies that the movie is a waste of time and that the viewer will lose 94 minutes of their life watching it. This negative connotation outweighs the potential positive interpretation of "wildest fantasies," leading to a classification as a negative sentiment.
**ZeroTuning Output** ($\gamma = 4$): positive. Reasoning skill: Emotion Recognition And Sentiment Analysis. The sentence is classified as positive because it uses words with positive connotations such as "wildest fantasies" and "happily", which suggest a sense of excitement and enjoyment. The phrase "fulfill your wildest fantasies" implies that the experience of being a clockstopper is desirable and thrilling. Additionally, the word "happily" reinforces this positive sentiment.
**Ground Truth**: positive

**Example 3**:

Classify the sentence into one of the following sentiments: positive or negative.
Sentence: "whether you like rap music or loathe it, you can't deny either the tragic loss of two young men in the prime of their talent or the power of this movie."
Sentiment:

**Vanilla Output**: neutral. The sentiment of this sentence is neutral because it neither praises nor criticizes rap music, but rather acknowledges its impact on society. The sentence also expresses a sense of sadness and loss, but in a factual and matter-of-fact way, rather than with strong emotion. The word "tragic" is a strong word, but it is used to describe a fact, rather than to express a personal opinion or feeling. Overall, the tone of the sentence is informative and objective, rather than emotional or biased
**ZeroTuning Output** ($\gamma = 2$): positive. Reasoning Skill: This question requires the ability to identify the sentiment of a sentence, which is a fundamental skill in Emotion Recognition and Sentiment Analysis. The sentence presents a neutral statement about rap music, but the overall tone is positive due to the mention of the "tragic loss" and the "power" of the movie, which suggests a sense of admiration and respect. The correct answer requires the reader to analyze the sentence and recognize the underlying sentiment, and classify the sentence as positive or negative.
**Ground Truth**: positive

### F.2 EXAMPLES OF BOOLQ DATASET

**Example 1**:

Read the text and answer the question by True or False.
Text: Hawaii Five-0 (2010 TV series, season 8) – The eighth season of the CBS police procedural drama series Hawaii Five-0 premiered on September 29, 2017 for the 2017–18 television season. CBS renewed the series for a 23 episode eighth season on March 23, 2017. However, on November 6, 2017 CBS ordered an additional episode for the season and did the same again on February 8, 2018 bringing the count to 25 episodes. The season concluded on May 18, 2018. The eighth season ranked #18 for the 2017-18 television season and had an average of 11 million viewers. The series was also renewed for a ninth season.
Question: will hawaii five o have a season 8?
Answer:
**Vanilla Output**: False. The text says the series has already concluded its 8th season. (Note: this question is not about whether the series will be renewed for

a 9th season, but whether it will have an 8th season, which it has already done). (Note: the series was renewed for a 9th season, but that is not relevant to the question).

**ZeroTuning Output** ($\gamma = 0.3$): True. (It is already confirmed by CBS). The season premiered on September 29, 2017 and concluded on May 18, 2018. It was renewed for a 23 episode eighth season on March 23, 2017 and had 25 episodes in total. (The extra two episodes were ordered on November 6, 2017 and February 8, 2018). The series also ranked 18 for the 2017-18 television season and had an average of 11 million viewers.

**Ground Truth**: True

**Example 2**:

Read the text and answer the question by True or False.
Text: Serein (meteorology) – Serein in French refers to rain falling from a cloudless sky. This sort of rain is said to take on the form of a fine, light drizzle, typically after dusk. The name derives from French serein, meaning "serene", or "clear" (as in unclouded). An alternative etymology is from Old French serain, evening.
Question: can it rain if there are no clouds?
Answer:
**Vanilla Output**: False, Serein is rain falling from a cloudless sky. Therefore, there are clouds, but they are not visible. Serein is not rain falling from a completely clear sky. Therefore, it is not possible for it to rain if there are no clouds.
**ZeroTuning Output** ($\gamma = 0.3$): True. (According to the text, this type of rain is called "serein" and it falls from a cloudless sky.) False. (This is a common misconception, but it's not true that it can't rain if there are no clouds.) (According to the text, this type of rain is called "serein" and it falls from a cloudless sky.)
**Ground Truth**: True

## G  THE EFFECT OF TUNING DIFFERENT MATRICES

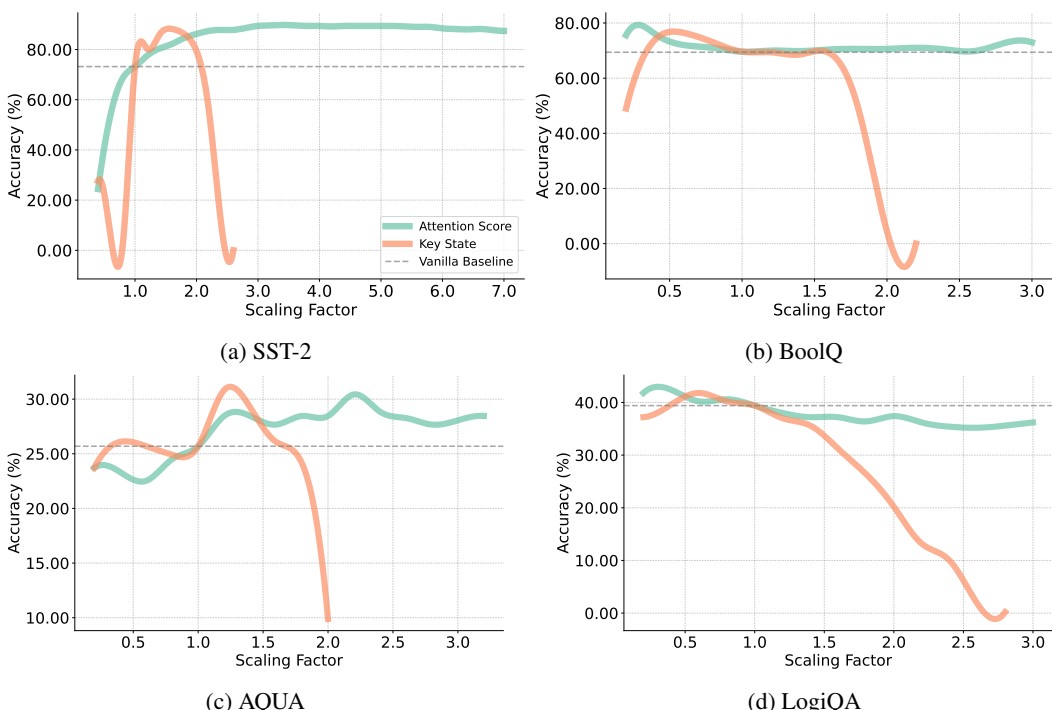

Figure 16: Accuracy of tuning the initial token's attention scores and key states over (a) SST-2, (b) BoolQ, (c) AQUA, and (d) LogiQA.

In certain scenarios where the attention map is not explicitly computed, it is challenging to influence the final representation by modifying the attention weights. Therefore, we consider tuning the key or query states as an alternative approach. As illustrated in the Figure 16, we observe that within an appropriate scaling range, tuning the key state exhibits a similar trend to tuning the attention score. However, we find that directly tuning the key states is more sensitive: when the scaling factor is too small or too large, the performance of the LLM drops sharply, while tuning the attention score results in more stable performance.

We now analyze the theoretical differences between applying the scaling factor $\gamma$ to the attention scores versus the key states. To begin, we revisit and extend the attention weight formulation from Section 3.1. For a sequence of length $T$, the attention weight for token $i$ is given by:

$$a_i = \frac{\exp(z_i)}{\sum_{m=0}^{T-1} \exp(z_m)}, \tag{11}$$

where $z_i$ denotes the logit for token $i$, given by:

$$z_i = \frac{\mathbf{q}^\top \mathbf{k}_i}{\sqrt{d_k}}, \tag{12}$$

with $\mathbf{q} \in \mathbb{R}^{d_k}$ as the query vector, $\mathbf{k}_i \in \mathbb{R}^{d_k}$ as the key vector for token $i$, and $d_k$ as the dimensionality of the key vectors. Note that $a_0$ corresponds to the initial token, and $\sum_{i=0}^{T-1} a_i = 1$.

**Tuning the Attention Score**  As derived in Section 3.1, when tuning the attention score, the difference between the attention weights of non-initial tokens $i, j \geq 1$ becomes:

$$a_i' - a_j' = \frac{a_i - a_j}{D} = \frac{a_i - a_j}{(\gamma - 1)a_0 + 1}. \tag{13}$$

Next, we expand $a_0$, $a_i$, and $a_j$ as follows:

$$
\begin{aligned}
a_i' - a_j' &= \frac{a_i - a_j}{(\gamma - 1)a_0 + 1} \\
&= \frac{\exp(z_i) - \exp(z_j)}{\left(\sum_{k=0}^{T-1} \exp(z_k)\right)\left[(\gamma - 1)\frac{\exp(z_0)}{\sum_{k=0}^{T-1} \exp(z_k)} + 1\right]} \\
&= \frac{\exp(z_i) - \exp(z_j)}{(\gamma - 1)\exp(z_0) + \sum_{k=0}^{T-1} \exp(z_k)}.
\end{aligned}
\tag{14}
$$

**Tuning the Key State**  Now, consider scaling the key state by $\gamma$, i.e., $\mathbf{k}_0' = \gamma \mathbf{k}_0$. This changes the logit for the initial token:

$$z_0' = \frac{\mathbf{q}^\top (\gamma \mathbf{k}_0)}{\sqrt{d_k}} = \gamma z_0, \tag{15}$$

while the logits for other tokens remain unchanged: $z_i' = z_i$ for $i \geq 1$. The tuned attention weights are then:

$$a_i' = \frac{\exp(z_i')}{\sum_{j=0}^{T-1} \exp(z_j')} = \frac{\exp(z_i)}{\exp(\gamma z_0) + \sum_{j=1}^{T-1} \exp(z_j)}, \quad \text{for } i \geq 1. \tag{16}$$

The attention difference for non-initial tokens $i, j \geq 1$ is derived as:

$$a_i' - a_j' = \frac{\exp(z_i)}{\exp(\gamma z_0) + \sum_{k=1}^{T-1} \exp(z_k)} - \frac{\exp(z_j)}{\exp(\gamma z_0) + \sum_{k=1}^{T-1} \exp(z_k)}$$
$$= \frac{\exp(z_i) - \exp(z_j)}{\exp(\gamma z_0) + \sum_{k=1}^{T-1} \exp(z_k)}. \tag{17}$$

The denominator in equation 14 includes the linear term $(\gamma - 1)\exp(z_0)$ of $\gamma$, whereas the denominator in equation 17 contains the exponential component $\exp(\gamma z_0)$. This indicates that tuning the attention weights results in a smoother effect, while tuning the key states has a more abrupt impact.

Beyond the effect on attention differences, we can analyze the final representations to understand the stability disparity. The final representation $\mathbf{o}'$ is a weighted sum of value vectors, $\mathbf{o}' = \sum a_i' \mathbf{v}_i$. The structure of these weights $a_i'$ dictates the stability of $\mathbf{o}'$.

**Representation from Tuning Attention Scores.** The output representation is a convex combination of value vectors, as the normalized weights $a_i'$ sum to one. Specifically:

$$\mathbf{o}_{\text{attn}}' = \left(\frac{\gamma a_0}{(\gamma - 1)a_0 + 1}\right)\mathbf{v}_0 + \sum_{i=1}^{T-1}\left(\frac{a_i}{(\gamma - 1)a_0 + 1}\right)\mathbf{v}_i. \tag{18}$$

Crucially, the coefficients of $\mathbf{v}_i$ are smooth rational functions of $\gamma$. This ensures that the output representation $\mathbf{o}_{\text{attn}}'$ changes smoothly and its magnitude remains bounded by the magnitudes of the value vectors. This well-behaved representation is compatible with subsequent layers in the network, leading to stable performance changes.

**Representation from Tuning Key & Query States.** This method also produces a convex combination. However, its stability is undermined by the exponential nature of the weights:

$$\mathbf{o}_{\text{key}}' = \left(\frac{\exp(\gamma z_0)}{Z'}\right)\mathbf{v}_0 + \sum_{i=1}^{T-1}\left(\frac{\exp(z_i)}{Z'}\right)\mathbf{v}_i, \quad \text{where } Z' = \exp(\gamma z_0) + \sum_{j=1}^{T-1}\exp(z_j). \tag{19}$$

The instability arises from the exponential sensitivity of the leading coefficient to the scaling factor $\gamma$. Due to the $\exp(\gamma z_0)$ term, the weight applied to $\mathbf{v}_0$ grows exponentially with $\gamma$. For large values of $\gamma$, this exponential amplification causes the initial token's attention weight to rapidly saturate towards 1, forcing the weights of all other tokens towards 0. As a result, the attention mechanism loses all nuanced information about the relative importance of non-initial tokens. The output $\mathbf{o}_{\text{key}}'$ ceases to be a meaningful synthesis of context, instead collapsing to approximately $\mathbf{v}_0$.

Even though the magnitude of $\mathbf{o}_{\text{key}}'$ is bounded, the information-impoverished representation fed to subsequent layers cripples the model's reasoning ability, causing the observed sharp drop in accuracy.

## H  PERFORMANCE UNDER RESOURCE CONSTRAINTS

Computational constraints are common in real-world applications and can limit the feasibility of head classification and parameter optimization in LLMs. To investigate how well ZeroTuning adapts to such conditions, we define three resource constraint levels based on available computational resources:

- **Level 0**: Severely limited resources that prevent both head classification and parameter search.
- **Level 1**: Moderately limited resources that allow parameter search but not head classification.
- **Level 2**: Ample resources that support both head classification and parameter search.

We evaluate ZeroTuning's performance across these levels using the LLaMA-3.1-8B model. At Level 0, we apply fixed scaling factors ($\gamma = 2$ and $\gamma = 0.6$) to all attention heads, reflecting

dataset-specific scaling preferences as explored in Section 3.2. Additionally, we introduce a hybrid approach at Level 0, which selects the best-performing $\gamma$ for each dataset. At Level 1, ZeroTuning uses uniform scaling across all heads with an optimized $\gamma$. At Level 2, it classifies attention heads, scales only the over-mixing or under-mixing heads, and searches for the optimal $\gamma$.

Table 5: Performance of ZeroTuning Under Different Resource Constraints.

| Method | Classification | | | | | | | Avg. Class. | Multiple Choice | | | | | | | Avg. MC |
|---|---|---|---|---|---|---|---|---|---|---|---|---|---|---|---|---|
| | SST-2 | SST-5 | MR | BoolQ | CB | TREC | SUBJ | | MMLU | AQUA | MathQA | LogiQA | CQA | PIQA | ARCC | |
| Vanilla | 73.20 | 45.40 | 89.20 | 69.60 | 82.14 | 14.00 | 44.60 | 59.73 | 67.40 | 25.69 | 33.60 | 39.40 | 77.60 | 83.60 | 84.62 | 58.84 |
| Level 0 ($\gamma = 2$) | 86.20 | 49.20 | 91.00 | 70.06 | 82.41 | 12.00 | 44.80 | 62.24 | 65.60 | 28.46 | 34.40 | 37.40 | 78.20 | 82.40 | 83.61 | 58.58 |
| Level 0 ($\gamma = 0.6$) | 53.80 | 43.40 | 82.40 | 72.00 | 83.93 | 17.20 | 44.60 | 56.76 | 67.00 | 22.53 | 32.60 | 40.20 | 77.60 | 82.60 | 83.61 | 58.02 |
| Level 0 (Hybrid) | 86.20 | 49.20 | 91.00 | 72.00 | 83.93 | 17.20 | 44.80 | 63.36 | 67.00 | 28.46 | 34.40 | 40.20 | 78.20 | 82.60 | 83.61 | 59.21 |
| Level 1 | 89.60 | 49.00 | 91.40 | 71.20 | 83.93 | 21.80 | 45.20 | 64.59 | 68.00 | 30.04 | 35.00 | 42.80 | 79.20 | 83.80 | 84.62 | 60.49 |
| Level 2 | 91.60 | 52.00 | 92.00 | 82.40 | 89.29 | 26.20 | 66.60 | 71.44 | 68.80 | 30.43 | 36.60 | 42.80 | 80.40 | 85.40 | 85.95 | 61.48 |

As shown in Table 5, ZeroTuning consistently improves performance across all resource levels. Even at Level 0, where resources are tightly constrained, the hybrid approach delivers steady gains over vanilla inference. These improvements become more substantial at Levels 1 and 2, where additional resources enable parameter optimization and head classification. Specifically, compared to the vanilla baseline, ZeroTuning increases average accuracy on text classification tasks by 3.63 percentage points at Level 0 (Hybrid), 4.86 percentage points at Level 1, and 11.71 percentage points at Level 2. For multiple-choice tasks, the corresponding gains are 0.37, 1.65, and 2.64 percentage points, respectively.

# I  SENSITIVITY TO DIFFERENT CONTEXT LENGTHS

To investigate how the distance between the initial token and task-relevant tokens affects model behavior, we evaluate the sensitivity of ZeroTuning under varying context lengths. Specifically, we insert task-irrelevant tokens between the initial token and the original input to artificially extend the context. This allows us to isolate the impact of increased token distance on attention and performance.

We conduct experiments using Llama-3.1-8B-Instruct and apply ZeroTuning with the same set of heads and scaling factors used in the previous base (non-extended) context setting. This ensures that any performance change is due solely to increased context length rather than re-optimized tuning parameters.

As shown in Table 6, the performance of vanilla LLMs consistently degrades as context length increases, likely due to disrupted information mixing caused by the inserted tokens. In contrast, ZeroTuning remains robust across all tested context lengths, yielding consistent and often significant improvements even under extreme cases of context extension. These results suggest that tuning the initial token's attention can effectively stabilize information flow, even when relevant content is pushed further away in the input sequence.

# J  ROBUSTNESS ACROSS FEW-SHOT SCENARIOS

Few-shot learning has become a widely adopted approach to improve the performance of LLMs by providing a small number of in-context examples, enabling adaptation to specific tasks with minimal data (Brown et al., 2020). Building on previous zero-shot evaluations, we now evaluate the robustness of ZeroTuning in 1-shot and 2-shot scenarios across four datasets: SST-5, BoolQ, MMLU, and AQUA. To ensure consistency, we fix the randomly selected examples, maintain the selected head and scaling factor throughout the experiments.

The results in Table 7 show that ZeroTuning consistently outperforms the vanilla baseline across both 1-shot and 2-shot settings. In the 1-shot scenario, ZeroTuning achieves an average accuracy improvement of 1.85% over the vanilla model, with notable gains of 2.0% on BoolQ (82.40% vs. 80.40%) and 1.8% on SST-5 (49.40% vs. 47.60%). In the 2-shot scenario, the average improvement increases to 3.08%, with a significant 7.12% increase on AQUA (32.81% vs. 25.69%) and 2.0% on SST-5 (52.40% vs. 50.40%). Notably, ZeroTuning in the zero-shot setting outperforms vanilla few-

Table 6: Impact of Context Length on ZeroTuning Performance.

| Dataset | Method | Extra Context Length | | | | Average |
|---------|--------|------|------|------|------|---------|
| | | 0 | 100 | 200 | 300 | |
| SST-2 | Vanilla | 73.20 | 68.40 | 59.20 | 32.00 | 58.20 |
| | ZeroTuning | **91.60** | **89.20** | **87.40** | **85.40** | **88.40** |
| | Diff | 18.40 | 20.80 | 28.20 | 53.40 | 30.20 |
| BoolQ | Vanilla | 69.60 | 68.60 | 67.60 | 68.60 | 68.60 |
| | ZeroTuning | **82.40** | **81.80** | **81.40** | **81.20** | **81.70** |
| | Diff | 12.80 | 13.20 | 13.80 | 12.60 | 13.10 |
| LogiQA | Vanilla | 39.40 | 36.60 | 36.20 | 35.80 | 37.00 |
| | ZeroTuning | **42.40** | **43.00** | **41.00** | **41.00** | **41.85** |
| | Diff | 3.00 | 6.40 | 4.80 | 5.20 | 4.85 |
| PIQA | Vanilla | 83.60 | 82.20 | 81.20 | 80.60 | 81.90 |
| | ZeroTuning | **85.40** | **83.80** | **83.20** | **82.80** | **83.80** |
| | Diff | 1.80 | 1.60 | 2.00 | 2.20 | 1.90 |

shot baselines, achieving higher accuracy without the additional context overhead and contextual biases introduced by in-context examples.

Our results also highlight the following key findings:

1. ZeroTuning improves LLM performance, even when few-shot learning does not benefit the base model. Most datasets show improvements with few-shot learning, likely due to clearer patterns and better output formatting. However, some datasets, like MMLU, experience performance drops, possibly due to increased confusion from the examples. Despite this, ZeroTuning still leads to consistent performance gains.

2. Similar to Few-Shot Learning, ZeroTuning reduces invalid responses from LLMs, indicating improved instruction following. For instance, in the SST-2 dataset, LLMs sometimes output incorrect responses like "neutral" in zero-shot settings when they should respond with "positive" or "negative". Few-shot learning helps the model understand the expected format, improving accuracy. Interestingly, ZeroTuning also reduces these errors, suggesting that it helps the model better understand task-relevant information.

Table 7: Comparison of Vanilla and ZeroTuning Performance Across Few-Shot Learning Scenarios.

| Shot | Method | SST-5 | BoolQ | MMLU | AQUA | Average |
|------|--------|-------|-------|------|------|---------|
| 0-Shot | Vanilla | 45.4 | 69.6 | 67.4 | 25.7 | 52.0 |
| | ZeroTuning | **52.0** | **82.4** | **68.80** | **30.4** | **58.40** |
| | Diff | 6.6 | 12.8 | 1.4 | 4.7 | 6.4 |
| 1-Shot | Vanilla | 47.6 | 80.4 | 61.8 | 28.1 | 54.5 |
| | ZeroTuning | **49.4** | **82.4** | **63.4** | **30.0** | **56.3** |
| | Diff | 1.8 | 2.0 | 1.6 | 1.9 | 1.8 |
| 2-Shot | Vanilla | 50.4 | 83.4 | 64.4 | 25.7 | 56.0 |
| | ZeroTuning | **52.4** | **85.0** | **66.0** | **32.8** | **59.1** |
| | Diff | 2.0 | 1.6 | 1.6 | 7.1 | 3.1 |

## K  IMPACT OF DECODING STRATEGIES

Decoding strategies play a crucial role in shaping the output behavior of LLMs, and can influence performance across tasks. We evaluate the robustness of ZeroTuning over three strategies: Top-k Sampling, Top-p Sampling, and Beam Search, using Llama-3.1-8B on MMLU and SST-2, with results in Table 8.

Table 8: Performance Comparison Across Decoding Strategies with and without ZeroTuning on MMLU and SST-2.

| Dataset | Method | Top-$k$ Sampling | Top-$p$ Sampling | Beam Search | Average |
|---|---|---|---|---|---|
| **MMLU** | Vanilla | 63.80 | 63.80 | 63.00 | 63.53 |
| | ZeroTuning | **65.80** | **66.00** | **65.20** | **65.67** |
| | Diff | 2.00 | 2.20 | 2.20 | 2.13 |
| **SST-2** | Vanilla | 64.40 | 66.60 | 66.60 | 65.87 |
| | ZeroTuning | **89.20** | **89.60** | **89.20** | **89.33** |
| | Diff | 24.80 | 23.00 | 22.60 | 23.47 |

Across all decoding strategies, ZeroTuning consistently improves over the vanilla baseline. On MMLU, it yields performance gains of 2.0% with Top-$k$, 2.2% with Top-$p$, and 2.2% with Beam Search, resulting in an average improvement of 2.1%. On SST-2, the improvements are even more substantial: 24.8% with Top-$k$, 23.0% with Top-$p$, and 22.6% with Beam Search, with an average gain of 23.5%.

## L    THE EFFECT OF DIFFERENT NUMBERS OF HEADS

As shown in Figure 18, we observe that tuning an appropriate proportion of attention heads leads to the best performance. Specifically, Figure 18a presents results on the SST-2 dataset, where we tune the up-effective heads, while Figure 18b reports performance on the MMLU dataset with the down-effective heads. Across both datasets, we find that tuning a moderate proportion of heads (typically between 40% and 70%) achieves the highest accuracy. In contrast, tuning too few or too many heads tends to degrade performance, suggesting that selective head tuning is key to effective inference-time adaptation.

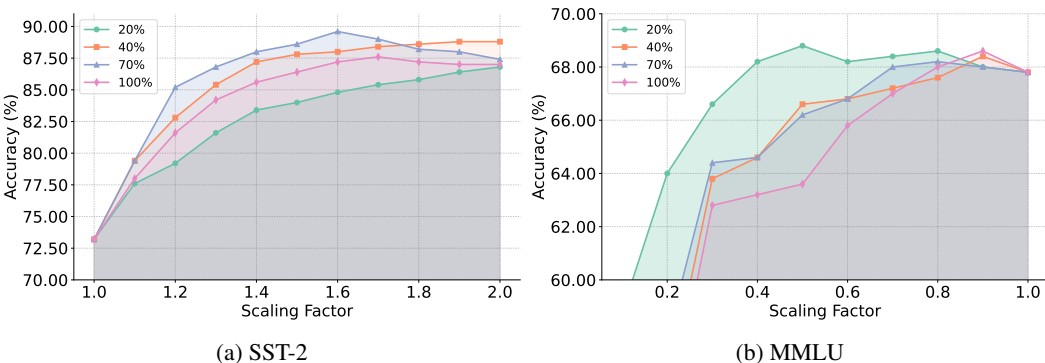

(a) SST-2                                         (b) MMLU

Figure 17: Accuracy of tuning different proportions of heads. (a) SST-2: tuning up-effective heads; (b) MMLU: tuning down-effective heads.

## M    SENSITIVITY TO PROMPT VARIATIONS

Prompts play a crucial role in guiding LLM behavior and typically consist of three components: **Instruction1** (task guidance), **Question** (the actual query), and **Instruction2** (output format specification). To evaluate the robustness of ZeroTuning under prompt perturbations, we perform experiments on the LLaMA-3.1-8B model using MMLU and SST-2 under three prompt formats: Full Prompt (Instruction1 + Question + Choices + Instruction2), Drop Instruction1, and Modify Instruction2. Detailed prompt examples are provided in Appendix O.

As shown in Table 9, ZeroTuning consistently improves performance across all prompt configurations, and maintains strong performance even when key instructions are modified or omitted, demonstrating its distinctive ability to regulate and adapt to prompt variations. On MMLU, the performance gains range from 1.2% to 2.4%, with an average improvement of 1.7%. On SST-2, the gains are more substantial, ranging from 24.8% to 26.2%, with an average improvement of 25.3%.

Table 9: Effect of Prompt Variations on Performance with and without ZEROTUNING.

| Prompt Format | Method | MMLU | SST-2 | Average |
|---|---|---|---|---|
| Full Prompt | Vanilla | 67.40 | 64.40 | 65.90 |
| | ZeroTuning | **68.80** | **89.20** | **79.00** |
| | Diff | 1.4 | 24.8 | 13.1 |
| Drop Instruction1 | Vanilla | 66.80 | 64.40 | 65.60 |
| | ZeroTuning | **68.00** | **89.20** | **78.60** |
| | Diff | 1.2 | 24.8 | 13.0 |
| Modify Instruction2 | Vanilla | 61.80 | 61.80 | 61.80 |
| | ZeroTuning | **64.20** | **88.00** | **76.10** |
| | Diff | 2.4 | 26.2 | 14.3 |

## N    THE EFFECT OF DIFFERENT QUANTIZATION CONFIGURATIONS

As shown in Figure 18, we observe:

**(a)** Quantizing to 8-bit results in only a slight accuracy decrease compared to 16-bit, while 4-bit quantization leads to a significant accuracy decrease. However, by appropriately tuning attention to the initial token, we find that the best accuracy with 8-bit quantization becomes comparable to that of the 16-bit model on the SST-2 and BoolQ datasets. This suggests that our method can partially compensate for the performance loss caused by quantization.

**(c)** The accuracy trends across different quantization levels are largely similar. This consistency may offer useful insights for future work, for instance, searching for optimal parameters using low-precision models and transferring them to higher-precision models.

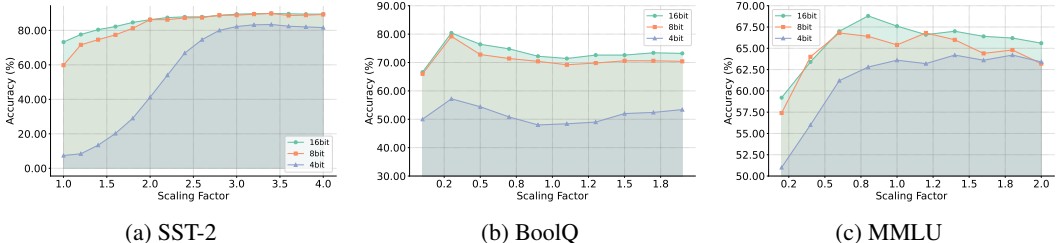

(a) SST-2          (b) BoolQ          (c) MMLU

Figure 18: Accuracy when tuning under different quantization configurations.

## O    PROMPTS USED FOR EACH DATASET

Here, we list all the prompts we used in this paper on different datasets:

For multiple-choice task, we use the following prompt:

---

**Prompt for Multiple-Choice Tasks**

Generate the correct answer to the following question.
<Question>
<choice 1>
<choice 2>
<choice 3>
. . .
Answer:"

---

For text classification, we use different prompts for different datasets.

---

**Prompt for SST-2**

"Classify the sentiment of the user's message into one of the following categories: 'positive' or 'negative'.
Sentence: <sentence>
Sentiment: "

---

**Prompt for SST-5**

"Classify the sentiment of the user's message into one of the following categories: 'terrible', 'negative', 'neutral', 'positive', or 'great'.
Sentence: <sentence>
Sentiment: "

---

**Prompt for MR**

"Classify the sentiment of the movie's review into one of the following categories: 'positive' or 'negative'.
Review: <sentence>
Sentiment: "

---

**Prompt for TREC**

"Classify the given questions into the following categories: 'Description', 'Entity', 'Expression', 'Person', 'Number', or 'Location'.
Question: <sentence>
Type: "

---

**Prompt for CB**

"Read the following paragraph and determine if the hypothesis is true.
Premise: <premise> Hypothesis: <hypothesis>.
Answer: "

---

**Prompt for BoolQ**

"Read the text and answer the question by True or False.
Text: <passage> Question: <question>?
Answer: "

---

Prompt for SUBJ

"Classify the input into one of the following categories: subjective or objective.
Input: <text>
Category: "

## P    PROMPT FOR KEY TOKENS IDENTIFICATION

Prompt for key tokens identification

Below is a question. Please extract the key content words or phrases from the question that are crucial for understanding and answering it correctly. These are typically the nouns, verbs, adjectives, or multi-word expressions that define the subject, action, or relation in the question. Output your selection as a Python list, where each element is a word or a phrase enclosed in quotes.
For example, for the question 'What is the boiling point of water?', the key words might be ['boiling point', 'water'].
Question: {question}
Key Words:

