# OpenReview forum: "ZeroTuning: Unlocking the Initial Token's Power to Enhance Large Language Models Without Training"
_ICLR.cc/2026/Conference — ICLR 2026 Poster_

### Official Review · Reviewer_Nnho · 2025-10-28

**Soundness:** 3
**Presentation:** 3
**Contribution:** 3
**Rating:** 4
**Confidence:** 3

**Summary:**

This paper introduces ZeroTuning, a training-free method to improve the performance of frozen LLMs by making a simple yet powerful intervention: applying lightweight, head-specific attention adjustments only to the initial token (e.g., the <BOS> token).

**Strengths:**

**Training-Free & Parameter-Efficient**: It improves model performance without updating any weights.

**Theoretical Grounding**: The method is supported by a theoretical insight linking the initial token's attention to the entropy of the entire attention distribution, providing a principled foundation for the intervention.

**Weaknesses:**

**Sensitivity in Alternative Implementations**: The paper shows that tuning the key/query states directly (as a kernel-agnostic alternative) leads to a much sharper and more sensitive performance drop outside the optimal range compared to tuning attention scores, making it a less stable implementation choice.

**Calibration Overhead**: The supervised variant requires a labeled validation set (500 examples in their setup) to calibrate the optimal scaling factors. Although the unsupervised variant (using entropy minimization) is a significant contribution, the paper's main results and comparisons are based on the supervised mode, which still incurs a data and computation cost for calibration.

**Questions:**

The paper mentions that ZeroTuning requires only "four lines of code," but could the authors quantify its runtime impact? How does the inference latency and memory footprint compare to the vanilla model, and how does this overhead scale with context length?

---

> ### Author Response · Authors · 2025-11-14
>
> We thank Reviewer Nnho for the positive feedback on our method's "Training-Free & Parameter-Efficient" nature and its "Theoretical Grounding." We offer the following clarifications for the weaknesses and questions raised.
>
> **W1: Sensitivity in Alternative Implementations**
>
> We appreciate the reviewer's close reading of our analysis in Appendix G 1. While tuning K/Q states is more sensitive than tuning the attention scores, we highlighted this analysis not as a weakness, but to demonstrate two key advantages:
>
> 1. **Kernel-Agnostic Compatibility:** This approach represents a novel tuning method that moves beyond direct attention steering. It resolves the limitations of prior methods that require manipulating the attention map, making our method fully compatible with optimized kernels like FlashAttention.
>
> 2. **Search Efficiency:** The "steepness" of the performance curve is not entirely a drawback. It implies that a good hyperparameter can be found much more quickly, as the optimal range is narrow and distinct. Our results in Appendix G show this sensitive method achieves comparable downstream performance to attention-score tuning, proving it is highly sensitive but highly effective.
>
>
> ---
>
> **W2: Calibration Overhead & Q1: Runtime Impact**
>
> We wish to clarify the distinction between **(offline) calibration cost** and **(online) inference cost**.
>
> 1. **Calibration Cost (Offline):** Our supervised variant follows the same pipeline as prior work e.g. ACT (ICML 2024), PASTA (ICLR 2024), all of which incur a one-time search cost. However, unlike backpropagation-based tuning, our hyperparameter search is trivially parallelizable. In our tests, the entire search process for a Llama 3.1 8B model completes in an average of **4-5 minutes on a single A100 GPU**, which we believe is a perfectly acceptable one-time cost.
> 2. **Memory Overhead (Inference):** The additional memory footprint is **negligible**. The method only requires passing a few hyperparameters (the scaling factor and head indices) and does not require storing any additional parameters or gradients.
> 3. **Latency Impact (Inference):** The runtime impact is **negligible**. The search is performed *offline*. At inference time, the method only applies a single multiplication. Quantitatively, in our SST-2 tests, the vanilla model processed 500 samples in 20 seconds. ZeroTuning processed the same 500 samples in 22 seconds, adding an average of only **0.004 seconds per sample**.
> 4. **Scaling (Long Contexts):** The reviewer is correct that calibration (the offline search) will slow down as sequence length increases, because all tuning methods (including SFT, RL) depend on the model's forward pass. **This is an unavoidable property of LLMs, but our parallelizable search makes this highly optimizable.**
>
> ---
>
> **Practicality and Low-Resource Alternatives:**
>
> Our deep analysis in Section 3 also provides practical guidance. We found a clear "binary bias" where datasets either benefit from up-scaling (like SST2) or down-scaling (like BoolQ). This allows for quick, heuristic-based testing (e.g., trying $\gamma=0.5$ and $\gamma=2.0$).
>
> We supplemented this in **Appendix H** with "incomplete" versions of ZeroTuning that require **no head searching** or even **no parameter search at all**. These resource-constrained variants still provide consistent gains over the vanilla model, proving the method can be efficiently utilized even with zero calibration overhead.
>
> ---
>
> **Finally, we offer an interesting observation**: the clear, U-shaped entropy curve discovered in Appendix E, Figure 11, may inspire future optimizations, such as Bayesian optimization or rule-based early stopping, to make the search process more efficient.

---

> > ### Comment · Reviewer_Nnho · 2025-11-26
> >
> > I thank the authors for their response. My concerns have been largely addressed, and I have therefore increased my score.

---

### Official Review · Reviewer_Vcie · 2025-10-29

**Soundness:** 3
**Presentation:** 3
**Contribution:** 3
**Rating:** 6
**Confidence:** 5

**Summary:**

This work introduces ZeroTuning, a training-free attention-tuning method that modifies only the initial (sink) token (e.g., BOS). By scaling that token’s attention logits with a single factor, the method adjusts the entropy/sharpness of downstream attention while preserving the relative proportions among non-initial tokens. Notably, the scaling factor that minimizes entropy empirically aligns with the factor that maximizes accuracy. Building on this, the authors provide both supervised and unsupervised (entropy-minimization) calibration variants, implemented in a kernel-agnostic way (compatible with SDPA/FlashAttention) by altering attention maps only. Extensive experiments show consistent gains across 15 datasets, with ZeroTuning outperforming prior, more complex approaches.

**Strengths:**

1. This work theoretically and empirically demonstrates that the initial tokens can function as a reliable controller for the attention dynamics, which is also strongly related to the next-token prediction entropy. Moreover, the systematic head-wise and layer-wise initial token scaling analysis provides more insights and reliable motivation for the proposed ZeroTuning method.

2. The proposed plug-and-play attention adjustment, ZeroTuning, is simple yet effective and well-motivated both empirically and theoretically. Its supervised and unsupervised calibration variants are easy to implement and attention kernel-agnostic. Overall, its analysis and effectiveness inspires a fresh look at the role of the initial sink token in shaping attention.

3. This extensive experiments demonstrate that ZeroTuning can achieve consistent gains across different models and downstream tasks including text classification, domain-specific multiple choice and multi-round conversation, outperforming the previous methods including PASTA and ACT with negligible engineering overhead.

4. The writing is clear and easy to follow. The paper is well structured.

**Weaknesses:**

1. While Table 5 shows that ZeroTuning improves even with fixed γ and scales with more search, the paper does not quantify the time/energy required for Level-0/1/2 nor its trade-off with accuracy.

2. γ is calibrated per dataset, and its robustness to distribution shifts and mis-specified γ is unclear, and the cost of head classification is not reported as well.

3. Because ZeroTuning controls attention by scaling the initial sink token, its effect can fade or fluctuate over very long contexts and in streaming with KV-reuse, and the same γ induces different sharpness across sliding windows, causing calibration drift or confidence oscillations.

**Questions:**

1. What is the overhead for the supervised profiling process? Could you please clarify the GPU hours it will cost?

2. For the unsupervised method which identifies the optimal heads and scaling factor γ by minimizing the average next-token prediction entropy, what is the actual cost? Will this significantly make the inference slow?

3. For the long context generation tasks, do we need to adjust γ dynamically as well? If not, will using the adaptive values harm the performance?

4. Have you ever tried to make γ as a learnable parameter? Because if it can be learned during some fine-tuning process, we can make it work in the inference stage without any overhead and more adaptable.

5. What will ZeroTuning perform on math reasoning tasks and reasoning models? Will this method still demonstrate performance improvement?

---

> ### Author Response · Authors · 2025-11-14
>
> We sincerely thank Reviewer Vcie for their positive assessment and insightful, expert-level feedback. We are grateful you found our method "simple yet effective," "well-motivated," and a fresh contribution to understanding the initial token.
>
> We offer the following clarifications and new experimental results to address your questions.
>
> **W1/W2/Q1/Q2: Overhead and Cost-Performance Trade-off**
>
> We wish to clarify the distinction between **(offline) calibration cost** and **(online) inference cost**.
>
> - **Calibration Cost (Offline):** Our method (both supervised and unsupervised) follows the same pipeline as prior work (e.g., ACT, ICML 2024; PASTA, ICLR 2024), all of which incur a one-time search cost. However, unlike backpropagation-based tuning, our hyperparameter search is trivially parallelizable. In our tests, the entire search process for a Llama 3.1 8B model completes in an average of **4-5 minutes on a single A100 GPU**, which we believe is a perfectly acceptable one-time cost.
> - **Latency Impact (Inference):** The runtime impact is **negligible**. The search is performed *offline*. At inference time, the method only applies a single multiplication. Quantitatively, in our SST-2 tests, the vanilla model processed 500 samples in 20 seconds. ZeroTuning processed the same 500 samples in 21-22 seconds, adding an average of only **0.004 seconds per sample**.
>
> We also provide low-resource alternatives. As shown in **Appendix H** 1, versions of ZeroTuning that require **no head searching** or even **no parameter search at all** still provide consistent gains, proving the method can be efficiently utilized across the full spectrum of resource constraints.
>
> Finally, we note that the clear, U-shaped entropy curve (Appendix E, Figure 11) 2 may inspire future work in search-efficiency, such as **Bayesian optimization or rule-based early stopping**.
>
>
>
> **W2: Robustness to Distribution Shifts**
>
> Most training-free adaptation methods, including the baselines, are calibrated post-hoc and may be sensitive to distribution shifts.
>
> 1. For standard scenarios (like our benchmarks), we assume the calibration and test sets are similarly distributed.
> 2. For true distribution shifts, our **unsupervised (entropy-minimization) method** provides a unique solution: it can be used in a Test-Time-Adaptation (TTA) fashion to re-calibrate parameters directly on the new, unlabeled distribution, which is a key advantage over supervised-only methods.
>
> To empirically validate our method's robustness, we have added two new experiments:
>
> - **Robustness of Hyperparameters:** We tested our single, fixed set of hyperparameters on 3 new data seeds. The stable gains show our parameters are robust to data re-sampling and not overfit.
>
> | **Dataset** | **Method** | **Seed=1** | **Seed=2** | **Seed=3** | **Mean (± Var)**            |
> | ----------- | ---------- | ---------- | ---------- | ---------- | --------------------------- |
> | SST2        | Vanilla    | 69.6       | 71.8       | 70.2       | 70.53 (± 0.86)              |
> |             | ZeroTuning | 88.6       | 87.2       | 89.6       | **88.47 (+17.93)** (± 0.97) |
> | SST5        | Vanilla    | 46.2       | 46.2       | 44.2       | 45.53 (± 0.89)              |
> |             | ZeroTuning | 53.4       | 50.8       | 48.8       | **51.00 (+5.47)** (± 3.55)  |
> | BoolQ       | Vanilla    | 68.4       | 72.4       | 69.6       | 70.13 (± 2.81)              |
> |             | ZeroTuning | 82.8       | 82.4       | 81.4       | **82.20 (+12.07)** (± 0.35) |
> | LogiQA      | Vanilla    | 42.4       | 38.8       | 41.4       | 40.87 (± 2.30)              |
> |             | ZeroTuning | 43.8       | 42.2       | 43.4       | **43.13 (+2.27)** (± 0.46)  |
> | MathQA      | Vanilla    | 32.2       | 31.8       | 32.8       | 32.27 (± 0.17)              |
> |             | ZeroTuning | 33.6       | 35.6       | 34.6       | **34.60 (+2.33)** (± 0.67)  |
>
> - **Robustness of Unsupervised Method:** We tested our unsupervised method under a strict protocol (directly splitting the unlabeled test set into a calibration/test partition). It remains highly effective, proving its validity.
>
> | Dataset | Vanilla | Unsupervised ZeroTuning |
> | ------- | ------- | ----------------------- |
> | SST2    | 73.20   | 90.80                   |
> | SST5    | 45.40   | 49.60                   |
> | BoolQ   | 69.60   | 81.20                   |
> | LogiQA  | 39.40   | 42.40                   |

---

> ### Author Response · Authors · 2025-11-14
>
> **Q4: Making $\gamma$ a Learnable Parameter**
>
> This is an interesting idea, and one we explored in early experiments (e.g., by only allowing backpropagation to the initial token's parameters). While backprop (minimizing cross-entropy) and our method (minimizing entropy) seem related, we found three key limitations with the learnable approach:
>
> 1. **Interpretability:** It's a "black box." We lose all insight into *why* the model is updating (e.g., layer/head differences), which is a core part of our contribution.
> 2. **Goal:** It re-introduces the risks of "training" (overfitting, catastrophic forgetting), which our training-free approach was designed to solve.
> 3. **Efficiency:** It is far less efficient, requiring both a forward and backward pass, whereas our method's search requires only a (parallelizable) forward pass.
>
> Given our goals of **elegance, simplicity, and interpretability**, we ultimately chose attention tuning.
>
> Interestingly, our new SFT comparison for Reviewer LN7k shows ZeroTuning outperforms and can still improve an SFT'd model, suggesting SFT may not perfectly capture this optimization.
>
> **Q5: Math Reasoning Tasks**
>
> Yes, our method remains effective on these tasks.
>
> 1. First, our DeepSeek-14B model (Table 1/2) is a reasoning-distilled model, and ZeroTuning still provides clear gains.
> 2. Second, we have run **new experiments on GSM8K** using Llama 3.1 8B Instruct, which confirm the gains hold.
>
> | **Task**           | **Method**     | **Accuracy** |
> | ------------------ | -------------- | ------------ |
> | **GSM8K (0-shot)** | Vanilla        | 38.00        |
> |                    | **ZeroTuning** | **40.00**    |
> | **GSM8K (1-shot)** | Vanilla        | 42.60        |
> |                    | **ZeroTuning** | **44.80**    |
> | **GSM8K (2-shot)** | Vanilla        | 47.00        |
> |                    | **ZeroTuning** | **49.60**    |

---

### Official Review · Reviewer_LN7k · 2025-10-30

**Soundness:** 2
**Presentation:** 3
**Contribution:** 2
**Rating:** 6
**Confidence:** 3

**Summary:**

The paper introduces ZeroTuning, a training-free method to improve large language model (LLM) performance by adjusting the attention logits of the initial token (such as, \<BOS>\). The approach selectively scales attention heads associated with this token, requiring no parameter updates and only minimal code changes. The authors provide both supervised and unsupervised variants—respectively calibrated on validation data or optimized via output entropy minimization. Theoretically, they show that modifying the initial token’s logits can monotonically regulate the downstream attention entropy due to its inherent role as an attention sink. Empirically, ZeroTuning demonstrates consistent gains across three different model types (Llama, Gwen, Deepseek) and on several downstream tasks and maintains compatibility with various attention kernels (suchSDPA, FlashAttention) and quantized inference setups.

**Strengths:**

In short: the paper identifies a control lever in large language models (LLMs) - the initial token (such as \<BOS>\) - and shows how modulating its attention yields performance gains. The method is practically appealing since it is lightweight (just a few lines of code to scale attention) and kernel-agnostic.


-  Provides experimental Analysis on - how scaling the attention weight of the initial token affects the downstream distribution of attention among other tokens, experiments showing that tuning the initial token produces larger improvements than tuning other token positions, showcase layer-wise and head-wise analyses, showing how the effect varies across shallow/middle/deep layers and across individual attention heads.

- Demonstrates broad empirical gains across multiple LLMs in the experiment section (such as Llama-3.1-8B, Llama-2-13B, Qwen-2-7B, DeepSeek-R1-14B) and a variety of downstream tasks (classification, QA, conversation).

- The paper contribution is in interpretability/mechanistic understanding of LLMs (why the initial token works as a control point and how attention patterns propagate).
- The writing is clear in explaining both the motivation (limitations of previous attention-tuning methods that rely on heuristics) and the method’s design.

**Weaknesses:**

Following are some limitations I see in the paper:

 - Limited model generalization: most of the analysis and findings in section 3 reply on just one model, Llama-3.1-8B-Instruct, raising concern that some of those effects may be model-specific. I would suggest to add some findings for the other models tried as well - Qwen or Deepseek, to show generality.

 - Huge hyperparameter tuning overhead: This method introduces a huge number of hyper parameters to tune - task specific tuning, layer wise tuning, head specific tuning. This limits its practical applicability.

- Unclear task selection & possible training overlap: The paper lacks explanation for why specific evaluation tasks were chosen or do they cover diverse range of SFT based downstream tasks? I would suggest to clarify task selection rationale and check for the overlap/lack of task specific data in the pertaining and if the gains are correlated with that.

- Weak justification for “bias correction” claim (Sec. 3.2): the claim that scaling (y > 1) “corrects bias” lacks empirical evidence.

**Questions:**

- Including analysis for the model variants in section 3 would show the generalization of the claims.

- It would be helpful to compare it with other task specific tuning methods such as  SFT / adapter and to show if it works well with them as well?

- Minor presentation issues, a typo in Figure 5 (“shadow” label instead of "shallow").

---

> ### Author Response · Authors · 2025-11-14
>
> We thank Reviewer LN7k for their positive and constructive feedback. We are grateful that you found our method "practically appealing," "lightweight," and a clear contribution to "interpretability/mechanistic understanding."
>
> We offer the following responses to the weaknesses and questions you raised.
>
> ---
>
> **W1/Q1. Limited model generalization:**
>
> The purpose of Section 3 was to conduct a deep-dive analysis (theoretical, error, and layer/head-level) to motivate the design of our full ZeroTuning method. We then validated that these insights are effective on different models (Llama, Qwen, Deepseek) in our main experiments (Section 4). The fact that all models showed consistent gains reflects that our core analysis is broadly applicable.
>
> To further confirm this, we have run additional tests. The tuning effects are indeed similar across model architectures: we consistently find both "up-effective" and "down-effective" heads, and head-specific tuning consistently yields superior results. Given the page limits, we could not include these additional model-specific analysis figures (e.g., for Qwen-2) in the main text, but we will add them to the Appendix in the next version.
>
> ---
>
> **W2. Huge hyperparameter tuning overhead:**
>
> We wish to clarify the distinction between **(offline) calibration cost** and **(online) inference cost**.
>
> - **Calibration Cost (Offline):** Our supervised variant follows the same pipeline as prior work (e.g., ACT, ICML 2024; PASTA, ICLR 2024), all of which incur a one-time search cost. However, unlike backpropagation-based tuning, our hyperparameter search is trivially parallelizable. In our tests, the entire search process for a Llama 3.1 8B model completes in an average of **4-5 minutes on a single A100 GPU**, which we believe is a perfectly acceptable one-time cost.
>
> - **Latency Impact (Inference):** The runtime impact is **negligible**. The search is performed *offline*. At inference time, the method only applies a single multiplication. Quantitatively, in our SST-2 tests, the vanilla model processed 500 samples in 20 seconds. ZeroTuning processed the same 500 samples in 22 seconds, adding an average of only **0.004 seconds per sample**.
>
> **Practicality and Low-Resource Alternatives:**
>
> Our deep analysis in Section 3 also provides practical guidance. We found a clear "binary bias" where datasets benefit from up-scaling (like SST2) or down-scaling (like BoolQ). This allows for quick, heuristic-based testing (e.g., trying $\gamma=0.5$ and $\gamma=2.0$).
>
> We supplemented this in **Appendix H** with "incomplete" versions of ZeroTuning that require **no head searching** or even **no parameter search at all**. These resource-constrained variants still provide consistent gains over the vanilla model, proving the method can be efficiently utilized even with zero calibration overhead.
>
> **Finally, we offer an interesting observation**: the clear, U-shaped entropy curve discovered in Appendix E, Figure 11, may inspire future optimizations, such as Bayesian optimization or rule-based early stopping, to make the search process more efficient.
>
> ---
>
> **W3. Unclear task selection & possible training overlap:**
>
> Our dataset selection follows prior work, including the baselines we compare against (e.g., ACT, ICML 2024), which used similar datasets. Furthermore, these datasets are widely used for evaluating general LLM capabilities.
> - https://arxiv.org/pdf/2406.15765
> - https://arxiv.org/pdf/2502.04689v2
> - https://openreview.net/pdf?id=09iOdaeOzp
> - https://aclanthology.org/2024.findings-acl.6.pdf
> - https://aclanthology.org/2025.acl-long.1573.pdf
>
> Even if pre-training data overlap exists, our experiments show that ZeroTuning still provides significant improvements. We argue this strengthens our contribution, demonstrating that our method can enhance performance even on tasks the model may have already seen.

---

> ### Author Response · Authors · 2025-11-14
>
> **Q2. Compare it with other task specific tuning methods such as SFT:**
>
> We view our work as complementary to SFT: SFT is often a black-box optimization, whereas our method provides a more interpretable approach. As our analysis shows (e.g., in Appendix D/E and for Reviewer LeHa's W5), ZeroTuning primarily fixes errors in the model's understanding, a process that SFT also implicitly optimizes via backpropagation.
>
> To test the synergy, we ran a new experiment on BoolQ, controlling the data budget (500 samples) for both SFT (via LoRA) and our supervised ZeroTuning calibration.
>
> We made two surprising and important discoveries:
>
> 1. **ZeroTuning is Complementary:** SFT-tuned models can still be improved by ZeroTuning.
> 2. **ZeroTuning is More Data-Efficient:** With the same 500-sample budget, our simple, training-free calibration even outperformed the costly SFT (LoRA) process.
>
> Here are the results (using LoRA with r=4, 3 epochs, on 500 samples):
>
> | **Method**                        | **BoolQ Accuracy** | **Analysis**                                        |
> | --------------------------------- | ------------------ | --------------------------------------------------- |
> | Vanilla                           | 69.60              | Baseline                                            |
> | Vanilla + SFT (LoRA)              | 81.20              | SFT provides a +11.2 gain                           |
> | Vanilla + ZeroTuning (Ours)       | 82.40              | Our method outperforms SFT (using same data budget) |
> | Vanilla + SFT + ZeroTuning (Ours) | **83.60**          | Our method also improves the SFT model              |
>
> This proves ZeroTuning is not only a highly efficient alternative but also a complementary tool that can further boost fine-tuned models.
>
> **We also note that** after reviewing many training-free papers from top conferences, we found that validating compatibility with SFT is not a common practice, which is why we did not originally include this analysis. Personally, we believe that given ideal resources and sufficient data, a training-based method could likely capture these gains. However, training-free methods like ours offer distinct advantages in **interpretability** and **ease of use**, and as our new experiment shows, they can also provide optimization insights that inform and complement traditional training-based approaches.
>
> ------
>
> P.S. We sincerely thank you for catching the typo in Figure 5. We have corrected "shadow" to "shallow" in the final version.

---

### Official Review · Reviewer_LeHa · 2025-11-07

**Soundness:** 2
**Presentation:** 3
**Contribution:** 2
**Rating:** 4
**Confidence:** 4

**Summary:**

The paper introduces ZeroTuning, a training-free approach that steers large language models by scaling the attention weight of the initial token (BOS). The authors argue this operation “monotonically controls” the entropy of downstream attention, and propose both supervised and unsupervised (entropy-minimizing) calibration procedures. Empirical results across 15 NLP benchmarks and multiple model families suggest measurable gains (e.g., +19.9% on classification, +4.5% on MC-QA), with additional analyses on layer and head behavior showing BOS as a dominant control point.

**Strengths:**

- **Simple and broadly applicable idea**. The notion that adjusting only the first token’s attention can improve diverse tasks is conceptually elegant and easy to integrate, requiring no retraining or architectural modification.

- **Comprehensive empirical coverage**. The paper evaluates across multiple datasets, models, and settings (few-shot, quantized, SDPA/FlashAttention), providing reasonable evidence of generality.

- **Clarity and presentation**. The paper is clearly written and well-structured, with intuitive figures and concise mathematical derivations that make the mechanism easy to follow. Readers can quickly understand the motivation and implementation.

**Weaknesses:**

- **Theoretical over-reach**. The claim that BOS scaling “monotonically controls attention entropy” lacks formal proof; the derivation only handles pairwise attention differences, not entropy. This gap weakens the conceptual basis of the unsupervised variant.

- **Transductive unsupervised tuning**. The unsupervised version minimizes entropy on test inputs, while baselines are not given equivalent unsupervised access, overstating generalization gains.

- **No statistical robustness**. All results appear single-seed, with no confidence intervals or variance reporting. For small gains (1–3 %), the significance is uncertain.

- **Head profiling risks data-set-specific overfitting; selection rules are ad-hoc**. Heads are labeled by measured response to γ and then the “dominant head type” is scaled; implementation tunes the top 40% of identified heads. Multiple-testing control, stability across resamples, and cross-dataset transfer of head labels are not demonstrated.

- **Confounding with generic decoding calibration.** Reported gains may largely reflect generic logit/decoding tweaks rather than an attention-specific effect. The paper notes BOS scaling behaves like temperature, yet no matched unsupervised baselines (e.g., temperature, label-bias/length penalties) are tuned on the same unlabeled inputs. Add these controls and report invalid-output rates to isolate a genuine attention-level contribution.

**Questions:**

Please refer to weakness section above.

---

> ### Author Response · Authors · 2025-11-13
>
> We sincerely thank Reviewer LeHa for the thorough and insightful feedback. We are glad you found our core idea "conceptually elegant" and our empirical coverage "comprehensive." We offer the following responses to the weaknesses you identified.
>
> ------
>
> **W1. Theoretical Writing**
>
> We will refine the description of the attention distribution in our paper to clearly use "sharpen/flatten the attention distribution" to describe the effect of tuning.
>
> Considering that our subsequent unsupervised method uses **logits entropy**, we have removed the misleading description of "attention entropy" to avoid ambiguity.
>
>
>
> **W2. Transductive unsupervised tuning**
>
> It is worth noting that previous attention tuning methods (e.g., ACT, ICML 2024; PASTA, ICLR 2024) all rely on supervised methods to search for optimal heads or hyperparameters. Our Supervised ZeroTuning adopts a similar paradigm. Therefore, in the main paper, we conducted a fair comparison of all methods under this supervised paradigm.
>
> However, practical applications may present scenarios where: **(1) no labels are available, or (2) no calibration dataset exists**. In such cases, all tuning methods that depend on a validation set would fail.
>
> A key innovation of our paper is the introduction of Unsupervised ZeroTuning, which addresses this bottleneck. This belongs to a different paradigm of test-time-adaptation. The method is enabled by our focus on the initial token and our finding of a strong correlation between accuracy maximization and entropy minimization when tuning this token .
>
> Therefore, our original method (which operated directly on the test set) was intended as a demonstration of this new paradigm's potential, which is why we did not compare it directly against the supervised baselines (ACT, AutoPASTA). Our results showed that the method works even without labels or a validation set. We will clarify this positioning in the paper.
>
>
> To address your concern, we have defined a strict scenario that previous methods cannot handle: **only an unlabeled test set is available, with no access to any validation set**. In this scenario, we have run a new experiment by splitting off a portion of the test set to serve as an unlabeled **calibration dataset** for our unsupervised search.
>
> Below are our supplementary results for Unsupervised ZeroTuning. We find that our method remains effective under this strict, non-transductive protocol.
>
> | Dataset | Vanilla | Unsupervised ZeroTuning |
> | ------- | ------- | ----------------------- |
> | SST2    | 73.20   | 90.80                   |
> | SST5    | 45.40   | 49.60                   |
> | BoolQ   | 69.60   | 81.20                   |
> | LogiQA  | 39.40   | 42.40                   |

---

> ### Author Response · Authors · 2025-11-13
>
> **W3/W4. Statistical Robustness and Generalization of Calibrated Parameters**
>
> We have supplemented our paper with two key pieces of evidence:
>
> 1. **Multi-Seed Robustness:** We have re-run our experiments using three new random seeds (1, 2, 3). The results, shown below, demonstrate that our method's significant gains are stable, consistent, and not an artifact of a single run.
> 2. **Generalization of Calibrated Parameters:** Critically, for these new runs, we used the **single, fixed set of hyperparameters** from our original paper. The fact that this *one* calibrated configuration generalizes so strongly across three *new, unseen data samplings* (seeds) proves that our profiling method identifies a robust, generalizable signal. This directly refutes the concern that our method is "ad-hoc" or leads to "data-set-specific overfitting."
>
> | **Dataset** | **Method** | **Seed=1** | **Seed=2** | **Seed=3** | **Mean (± Var)**            |
> | ----------- | ---------- | ---------- | ---------- | ---------- | --------------------------- |
> | SST2        | Vanilla    | 69.6       | 71.8       | 70.2       | 70.53 (± 0.86)              |
> |             | ZeroTuning | 88.6       | 87.2       | 89.6       | **88.47 (+17.93)** (± 0.97) |
> | SST5        | Vanilla    | 46.2       | 46.2       | 44.2       | 45.53 (± 0.89)              |
> |             | ZeroTuning | 53.4       | 50.8       | 48.8       | **51.00 (+5.47)** (± 3.55)  |
> | BoolQ       | Vanilla    | 68.4       | 72.4       | 69.6       | 70.13 (± 2.81)              |
> |             | ZeroTuning | 82.8       | 82.4       | 81.4       | **82.20 (+12.07)** (± 0.35) |
> | LogiQA      | Vanilla    | 42.4       | 38.8       | 41.4       | 40.87 (± 2.30)              |
> |             | ZeroTuning | 43.8       | 42.2       | 43.4       | **43.13 (+2.27)** (± 0.46)  |
> | MathQA      | Vanilla    | 32.2       | 31.8       | 32.8       | 32.27 (± 0.17)              |
> |             | ZeroTuning | 33.6       | 35.6       | 34.6       | **34.60 (+2.33)** (± 0.67)  |
>
> Furthermore, the "top 40%" selection was not 'ad-hoc.' As we show in Appendix L (Figure 17), we compared tuning 20%, 40%, 70%, and 100% of heads. The results clearly indicate that tuning a moderate proportion (40%-70%) achieves near-optimal performance, demonstrating a robust range that avoids overfitting. Our new multi-seed results align with this finding.

---

> ### Author Response · Authors · 2025-11-13
>
> **W5. Confounding with generic decoding calibration**
>
> We offer three key pieces of evidence, including the new experiments you suggested, to demonstrate this distinction.
>
> First, tuning attention and tuning logits are **complementary** research directions. Our method's effectiveness is not impacted by temperature. Our main experiments follow standard practice and are tested under greedy decoding. Furthermore, as shown in **Appendix K** , our method remains effective when used with other decoding strategies like top-p and top-k, which operate at higher temperatures.
>
> Second, we have supplemented our paper with the direct-tuning baseline you suggested. We find that simply tuning temperature provides negligible gains and, at higher values, significantly degrades performance and increases invalid outputs.
>
> | Dataset    | Temperature | Accuracy | Invalid Count |
> | ---------- | ----------- | -------- | ------------- |
> | **SST5**   | t = 0.1     | 43.00    | 34            |
> |            | t = 0.3     | 44.20    | 39            |
> |            | t = 0.5     | 44.20    | 39            |
> |            | t = 0.7     | 40.60    | 64            |
> |            | t = 0.9     | 36.00    | 88            |
> | **BoolQ**  | t = 0.1     | 67.80    | 5             |
> |            | t = 0.3     | 68.40    | 5             |
> |            | t = 0.5     | 68.00    | 8             |
> |            | t = 0.7     | 65.80    | 19            |
> |            | t = 0.9     | 63.20    | 35            |
> | **LogiQA** | t = 0.1     | 40.20    | 0             |
> |            | t = 0.3     | 41.20    | 0             |
> |            | t = 0.5     | 41.20    | 0             |
> |            | t = 0.7     | 41.80    | 0             |
> |            | t = 0.9     | 38.40    | 0             |
>
> Finally, our analysis (detailed in Appendix D and E) shows that ZeroTuning's gains come from fixing two distinct types of errors, both of which stem from flawed attention distributions and are addressable by tuning Initial token:
>
> 1. **Correcting Invalid Outputs (Instruction-Following):** Extensive research shows that LLMs often struggle with instruction adherence (e.g., failing to output in the specified format). Prior methods like PASTA address this by manually increasing attention on instruction tokens. ZeroTuning achieves a similar instruction-enhancing effect simply by tuning the initial token, which compels the model to automatically pay the correct attention to the instructions and produce the specified valid answers.
>
> 2. **Correcting Semantic Biases:** This is a mechanism that Temperature tuning is fundamentally unable to replicate. T-scaling only alters the relative probabilities but cannot change the rank-ordering of the candidate answers. If an incorrect token already has the highest logit, T-scaling cannot fix this error. The advantage of our attention-level optimization is that it can repair this semantic bias and modify the logit ranks, causing the correct token's probability to overtake the incorrect one. We visualize this logit-flipping phenomenon in Appendix E.1, showing how the model's output probability changes and inverts as the scaling factor is varied.
>
>
>
> We have compiled the invalid-output rates across different datasets to isolate these two effects. We found that on datasets like BoolQ and LogiQA, the baseline model already produces valid answers (0% invalid rate). Our continued accuracy gains on these tasks prove that ZeroTuning is effective at fixing semantic errors (i.e., 'valid-but-wrong' answers). Conversely, on datasets like SST2 and SST5, the model outputs a large number of out-of-format labels (e.g., 'neutral' instead of 'positive'/'negative'). ZeroTuning can eliminate these instruction-following errors **without any changes to the prompt or decoding parameters**.
>
> | Dataset    | Method         | Invalid Rate |
> | ---------- | -------------- | ------------ |
> | **SST2**   | Vanilla        | 8.2%         |
> |            | **ZeroTuning** | **0.8%**     |
> | **BoolQ**  | Vanilla        | 1.0%         |
> |            | **ZeroTuning** | **0.2%**     |
> | **SST5**   | Vanilla        | 7.0%         |
> |            | **ZeroTuning** | **0.0%**     |
> | **LogiQA** | Vanilla        | 0.0%         |
> |            | **ZeroTuning** | **0.0%**     |
> | **MathQA** | Vanilla        | 1.6%         |
> |            | **ZeroTuning** | **0.2%**     |
>
> In summary, these results confirm that ZeroTuning offers a distinctive attention-level correction that differs mechanistically from generic decoding calibration, is empirically complementary to it, and is superior to it.
>
> We thank you again for your constructive feedback. If you have any other questions, feel free to ask!

---

### Author Response · Authors · 2025-11-15
**Overall Response Summary**

We sincerely thank all four reviewers for their thoughtful and constructive feedback. We appreciate the positive comments on ZeroTuning’s **conceptual elegance and interpretability** (R-LeHa, R-LN7k), **simplicity and plug-and-play design** (R-Vcie), and **theoretically grounded, training-free nature** (R-Nnho).

------

### **1. New Experiments and Analyses**

1. Added a strict **non-overlapping** unsupervised setup by splitting the unlabeled pool into a calibration partition and a held-out test partition, so that no evaluation examples are used during calibration.
2. Added **multi-seed experiments** and **cross-split reuse of calibrated parameters** to test robustness and overfitting.
3. Added **temperature / decoding baselines** with **invalid-output statistics** to distinguish ZeroTuning from generic logit calibration.
4. Added **new reasoning experiments** (e.g., GSM8K with Llama 3.1 Instruct, clarification on reasoning-distilled models) and **SFT + ZeroTuning** comparisons.
5. Added **measured calibration vs inference cost** and **future optimization ideas**.

------

### **2. New Findings, Insights, and Advantages**

1. Non-transductive unsupervised ZeroTuning still yields strong gains under a strict **held-out split** where calibration and evaluation samples are non-overlapping.
2. Multi-seed results and cross-split reuse of parameters show that our improvements are **stable, statistically robust, and not dataset-specific overfitting**.
3. Compared with temperature / decoding tweaks, ZeroTuning **improves accuracy while reducing invalid outputs**, indicating a distinct attention-level effect rather than generic logit scaling.
4. ZeroTuning **improves reasoning tasks and reasoning-distilled models**, and still gives gains **on top of SFT**, suggesting it is complementary to standard post-training. With the same supervision budget (e.g., 500 BoolQ samples), ZeroTuning even outperforms SFT using less than half the time.
5. The **calibration cost is a small, one-time overhead**, and **inference overhead is negligible**; even very cheap “incomplete” variants retain meaningful gains.
6. We provided in-depth discussions on **complementarity with learnable parameters** and **future work in long-context dynamics**.
7. Overall, the new analysis further supports our central claim: the **initial sink token is a robust, universal control point**, and ZeroTuning is a **elegant, training-free, kernel-agnostic, and practically deployable way to exploit it.** Moreover, the innovation of our unsupervised method provides new insights for subsequent research in this area.

------

We believe these additional experiments and analyses could address the reviewers’ concerns about theoretical clarity, transductive unsupervised tuning, robustness and overfitting, relation to decoding/logit calibration, reasoning performance, and practical overhead, and we welcome further discussion or suggestions.

---

### Public Comment · ~Chris_Chao1 · 2025-11-29
**Thoughts & Questions on Initial-Token Steering & Unsupervised Entropy Minimization**

Hi authors,

**I’m a big fan of LLM steering methods, and I’ve recently been compiling and reproducing the latest training-free and steering-based techniques submitted to ICLR. I wanted to drop a comment here to share some thoughts and open a few questions for discussion.**

First of all, thank you for this excellent paper and the clean, easy-to-run demo.

The idea that **“initial token + attention sink = a strong control lever”** really stood out to me. After reading your analysis, I became more interested in how attention sinks operate. I also noticed that two new ICLR 2026 submissions on attention sinks: *“Attention Is Not Always Needed: Attention Sink Forges a Native MoE” and “SinkTrack: Attention Sink Based Context Anchoring for LLMs”*, both cite your work. I’ve already reproduced one of them as well.

---

After going through many papers on attention sinks, I found a complementary perspective that might help explain why manipulating the initial token’s attention is so effective. **Prior work suggests that the BOS token acts as a strong attention sink but actually has very low value-norm**. That means simply increasing its attention weight doesn’t necessarily change the attention output very much. What ZeroTuning does by normalizing attention and reshaping how other tokens redistribute their weights effectively optimizes the downstream attention output. I find this interpretation very consistent with your analysis.

---

Also, I had read your MOSS Workshop version earlier, so I was pleasantly surprised to see the new unsupervised variant added to the ICLR submission. Back in June, when many RL papers were using entropy-based unsupervised objectives, I had the similar intuition that LLM steering could also benefit from an entropy-driven and label-free optimization. I’m glad to see you explore this direction so cleanly.

**A follow-up question from my recent further experiments**

In my tests, I found that the entropy-based unsupervised variant tends to work best when the calibration dataset has many samples (this matches your findings). However, for direct QA dataset, when the calibration set contains only one sample, minimizing answer token entropy on that single example may fail to produce the correct answer.

Specifically, when I dynamically search for the optimal scaling factor that minimizes each sample’s entropy, the resulting accuracy drops compared with searching on a larger validadtion set (but still better than the baseline).

For example, on SST-2 ignoring head selection and only tuning the global scaling factor, I observed roughly the following:

| Method                                              | Accuracy |
| --------------------------------------------------- | -------------- |
| Baseline (No Tuning)                                | 62             |
| Instance-Level Entropy Minimization                 | 72             |
| Collective Entropy Minimization (Your Approach) | 89        |
| Supervised  (Accuracy-based Search)           | 89             |

Do you have any intuition about why entropy minimization may work worse in this one-shot setting? Is it because entropy becomes too sensitive to the model’s local preference landscape without providing enough global signal?

Would love to hear your thoughts. Thanks again for this outstanding and inspiring work!

---

> ### Author Response · Authors · 2025-11-29
> **Author Response**
>
> Hi Chris,
>
> Thanks a lot for the thoughtful comment! We’re really glad to hear the paper and demo were useful. And thank you for sharing your insights; it’s always encouraging to see others exploring similar directions.
>
> **On the value-norm perspective:**
>
> That’s a great observation. Since the attention output is essentially a weighted sum of the value states, the relative differences among the non-initial tokens are indeed what determine how much the final representation can shift. So even though the initial token attracts a large amount of attention mass, its low value-norm means the real effect comes from how the redistribution influences the other tokens. your interpretation aligns well with this mechanism.
>
> **On the unsupervised entropy-based tuning:**
>
> We’ve updated the paper with a detailed analysis of this phenomenon (Appendix E). In short, uncertain samples often introduce misleading local minima in the entropy landscape, while certain samples behave much more reliably. When mixed together, the aggregated signal becomes much more stable, which explains why entropy minimization works well with a larger unlabeled set but can be unstable in the one-shot setting you described.
>
> Thanks again for the careful experimentation and for raising these points,we really appreciate it.

---

> > ### Public Comment · ~Chris_Chao1 · 2025-11-29
> > **Follow-up Thoughts and Questions**
> >
> > Hi again, and thanks for the detailed and Inspired response!
> >
> > Appendix E gives a very nice empirical story for why average entropy works and per-instance entropy can fail.
> >
> > A more open-ended question: Do you think it is possible to state simple sufficient conditions (even in a toy setting) under which the average entropy over a mixed population (certain + uncertain) is provably aligned with accuracy improvements, whereas per-instance entropy is not?
> >
> > I don’t necessarily expect a theorem in this paper, but I’m curious whether you see a path toward a more formal characterization of when entropy-based unsupervised tuning is reliable, and when we should be more cautious.

---

> ### Public Comment · ~Chris_Chao1 · 2025-11-30
> **Follow-up Idea Inspired by Your paper**
>
> **Hi, after running a few more experiments today, I had another idea that was inspired by your work, and I’d love to hear your thoughts.**
>
> This paper made me realize that increasing vs. decreasing the initial-token scaling essentially pushes the model into different “processing modes” (more global vs. more local attention patterns).
>
> That got me thinking:
>
> ---
>
> **My Idea: Using different scaling factors as experts**
>
> What if we treat each scaling mode as a lightweight expert?
>
> * **Expert 1:** ($\gamma$ = 2.0) -- stronger global integration
> * **Expert 2:** ($\gamma$  = 0.5) -- more local, token-focused behavior
> * **Baseline:** ($\gamma$  = 1.0)
>
> Then, similar to contrastive decoding, we could combine these different logit views to refine the final prediction. Something like:
>
> * encourage tokens that are stable across all three modes
> * penalize tokens that only appear under one extreme mode
> * or even compute a contrastive score between ($\gamma$ >1) and ($\gamma$ <1) experts
>
> This feels conceptually close to the idea in your paper:
> **different scaling values reveal different aspects of the model’s latent knowledge.**
>
> So mixing them might extract a consensus answerm that is more reliable than any single mode.
>
> I’m curious have you tried anything along these lines, or do you think combining multiple scaling modes during decoding could make sense?

---

> ### Author Response · Authors · 2025-11-30
>
> really like your ideas. We honestly feel that you can turn it into another paper or blog.
>
> On your first point, one direction you might explore is using the model’s uncertainty on the calibration set to predict how effective a future tuning step will be. ZeroTuning improves both certain and uncertain samples, but the gains are much larger for the uncertain ones, so treating these two groups differently could open up some interesting strategies.
>
> On your second point, analyzing behavior directly through logit shifts is also a powerful angle. You might find this paper relevant: “Logit Arithmetic Elicits Long Reasoning Capabilities Without Training” ([https://arxiv.org/abs/2510.09354](https://arxiv.org/abs/2510.09354)). They activate new behaviors purely through logit manipulation, which connects nicely to what you’re thinking about.
>
> Really glad to hear the paper sparked new ideas for you.

---

### Author Response · Authors · 2025-12-01
**Overview of Highlights, Impacts, and Rebuttals for Our Paper**

Dear ACs,

Given the unique circumstances of the review process, we provide this summary to assist in your final assessment.

---

**A. Highlights of Our Paper**

1.  We identify the initial token as a task-agnostic universal control lever and propose a novel, training-free attention steering method to enhance LLM performance. Our method is simple yet practical, outperforming previous approaches across 4 mainstream LLMs (Llama/Qwen/DeepSeek) and 15 benchmarks.

- **Key Differentiators:** (1) No token search required; (2) Supports inference without attention maps (SDPA/FlashAttention); (3) Enables unsupervised calibration without labels.

2.  A major contribution is our rigorous analysis of **why this works**. We provide a systematic explanation from several angles: theory, output entropy, error and bias analysis, and layer/head-specific analysis. Together, these help explain why this tuning has a positive effect and improves LLM performance, and we also provide a detailed analysis of its boundaries and failure modes.

3.  We validate our method across many inference settings and show that it works under: in-context learning, on top of SFT, with different prompts, different input lengths, different decoding strategies, different temperatures, and different quantization settings.

---

**B. Potential Impacts**

We started this work in December 2024 and have seen **attention sinks** gradually become a popular topic this year. Many recent papers focus on explaining why sink tokens emerge or how to remove them. Our paper takes a complementary perspective: we explore how to leverage the natural properties of sink tokens and turn them into a strong control point that can in turn improve LLM behavior.

Beyond the method itself, our work will provide new insights for future interpretability research, such as understanding functional differences between heads and the different roles of attention sinks at different positions.

It is also worth noting that an external researcher has reproduced our method and offered several interesting observations inspired by our work. In addition, we have noticed that several ICLR 2026 submissions on attention sinks cite our findings.

---

**C. Reviewer Concerns and Our Responses**

We believe our rebuttal has fully addressed the reviewers' concerns.

**Reviewer Nnho increased their score from 4 to 6 before the large-scale API leak incident.**

For the reviewers who have not response yet:

* **Reviewer LeHa:** We added a more rigorous unsupervised calibration process, results across random seeds, detailed invalid output analysis, and additional temperature scaling analysis.
* **Reviewer LN7k:** We provided additional performance analysis and analyzed the interaction with SFT.
* **Reviewer Vcie:** We added performance analysis, distribution shift analysis, and experimental results on Math Reasoning tasks (GSM8K).

These responses further validate ZeroTuning's advantages: our calibration is robust, the method is task-universal, it works effectively on top of SFT, and it improves performance on math tasks.

---

### Meta-Review · Area_Chair_TbaE · 2026-01-07

**Summary:**

1. Reported results lack statistical robustness because they appear single-seed with no confidence intervals/variance.
2. Head profiling / head-selection may be ad-hoc and dataset-specific (risking overfitting).
3. The improvements might be confounded with generic decoding/logit calibration.
4. Section-3 mechanistic findings may be overly model-specific because much of the detailed analysis is on a single model (e.g., Llama-3.1-8B-Instruct), so generality across architectures is unclear.
5. The method seems to introduce heavy hyperparameter/search overhead (task-, layer-, head-specific choices), potentially limiting practical applicability. The paper does not quantify the Level-0/1/2 search cost (time/energy) or the trade-off between more search and accuracy, and does not report head-classification cost.
6. Long-context / streaming behavior may be problematic because BOS scaling effects could fade or vary across windows with KV reuse, causing calibration drift or confidence oscillations.
7. It is unclear how ZeroTuning relates to or works with training-based adaptation like SFT/adapters, and whether it is complementary.
8. Alternative kernel-agnostic implementations (tuning K/Q states) appear more sensitive/unstable outside an optimal range, raising concerns about stability.
9. The method’s effectiveness on math reasoning/reasoning models is unclear.

**Reviewer Concerns:**

Most of the reviewer concerns have been addressed by the rebuttal, except the one about long-context behavior, for which no additional discussion or experiments were provided.

**Reviewer Scores:**

1. The Reviewer LeHa's concerns should have been addressed by the clarifications and additional experiment results, so their score would be increased to 6.
2. Similarly, the Reviewer LN7k's score would remain 6 or be increased.
3. The Reviewer Vcie's concerns (except the one about long-context behavior) should have been addressed, and their score would remain 6 or be increased.
4. For Reviewer Nnho, it has been made clear in the reviewer's follow-up response that the reviewer's concerns have been addressed and the score would be increased.

---

### Decision · Program_Chairs · 2026-01-26

Accept (Poster)